# Visually Guided Decoding: Gradient-Free Hard Prompt Inversion with Language Models

**Donghoon Kim[1], Minji Bae[1], Kyuhong Shim[2]\*, Byonghyo Shim[1]\***
[1]Seoul National University [2]Sungkyunkwan University
{dhkim, mjbae, bshim}@islab.snu.ac.kr; khshim@skku.edu

## Abstract

Text-to-image generative models like DALL-E and Stable Diffusion have revolutionized visual content creation across various applications, including advertising, personalized media, and design prototyping. However, crafting effective textual prompts to guide these models remains challenging, often requiring extensive trial and error. Existing prompt inversion approaches, such as soft and hard prompt techniques, are not so effective due to the limited interpretability and incoherent prompt generation. To address these issues, we propose Visually Guided Decoding (VGD), a gradient-free approach that leverages large language models (LLMs) and CLIP-based guidance to generate coherent and semantically aligned prompts. In essence, VGD utilizes the robust text generation capabilities of LLMs to produce human-readable prompts. Further, by employing CLIP scores to ensure alignment with user-specified visual concepts, VGD enhances the interpretability, generalization, and flexibility of prompt generation without the need for additional training. Our experiments demonstrate that VGD outperforms existing prompt inversion techniques in generating understandable and contextually relevant prompts, facilitating more intuitive and controllable interactions with text-to-image models.

## 1 Introduction

In recent years, image generative models such as DALL-E and Stable Diffusion have shown remarkable success in generating high-fidelity images (Ramesh et al., 2022; Rombach et al., 2022; Podell et al., 2024). These models are widely used in a variety of applications, including visual content generation (*e.g.*, advertisement, movie, game), personalized content generation (*e.g.*, caricature, photo editing), and prototyping (*e.g.*, architecture and product design). Recent studies have shown that, just as humans can draw an image of an object solely based on detailed descriptions (*e.g.*, criminal composite sketch), generative models can generate images of objects using a well-crafted prompt, even if they have not been trained on those specific objects (Gal et al., 2023; Everaert et al., 2023).

A well-known drawback of these approaches is the difficulty in finding a textual description (*a.k.a.,* prompt) that effectively guides the generation of the desired visual content (Hao et al., 2024). For example, to create a culinary masterpiece (see Fig. 1), one needs to have enough knowledge on exquisite cooking styles such as French cuisine or molecular gastronomy. Without the expertise to craft prompts, users have to rely on laborious trial and error to generate high-quality gourmet images.

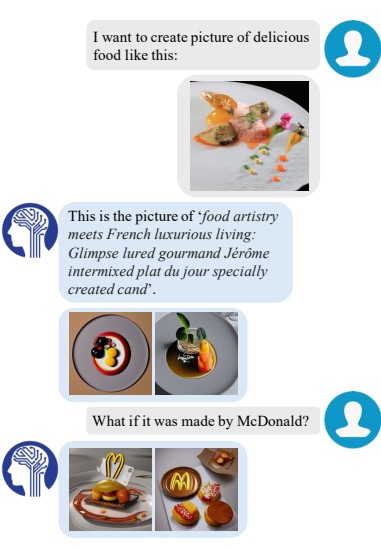

Figure 1: Visually Guided Decoding (**VGD**) works with any LLM without extra training, making it easy to integrate into a chat-based interface that offers interpretable and controllable text-to-image generation.

---

*Corresponding authors

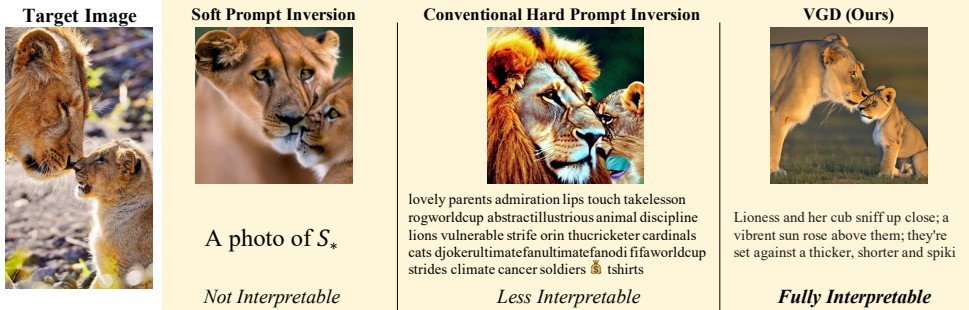

Figure 2: VGD generates fully interpretable prompts that enhance generalizability across tasks and models in text-to-image generation.

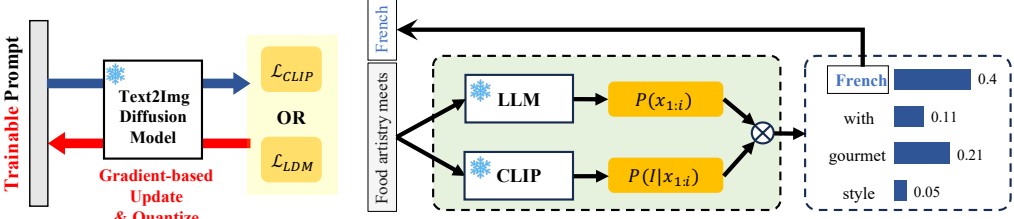

(a) Training-based Prompt Inversion     (b) Gradient-free, Visually Guided Hard Prompt Inversion (**Ours**)

Figure 3: While conventional prompt inversion techniques update prompt embeddings through gradient-based optimization and quantization, VGD is a gradient-free technique that utilizes large language models and CLIP to generate relevant sentences.

To address the difficulty in finding the prompt, soft prompt inversion techniques have been proposed (Gal et al., 2023; Kumari et al., 2023; Ruiz et al., 2023; Voynov et al., 2023). In these approaches, similar to the prompt learning technique in natural language processing (Li & Liang, 2021; Lester et al., 2021; Liu et al., 2022; Kim et al., 2024), an image is transformed into the embedding vector. Specifically, a learnable input vector for the conditioning network (Radford et al., 2021) in Stable Diffusion is trained to reconstruct user-provided images through an iterative de-noising process (*a.k.a.*, reverse diffusion process). After training, the learned vector is used as a soft prompt. Since the soft prompt is used as a word, to generate the desired image, it should be combined with other words (see Fig. 2). Unfortunately, it is a non-human-readable vector, users have no clue to handle it.

Recently, to mitigate the lack of interpretability, hard prompt inversion has been suggested (Mahajan et al., 2024; Wen et al., 2024). Essentially, training process of the hard prompt inversion is similar to the soft-prompt learning but the trained input vector is projected to the nearest neighbor in a predefined vocabulary to ensure human-readability (see Fig. 3 (a)). The potential drawbacks of hard prompt inversion are: 1) the vanishing gradient problem caused by the long gradient path in the reverse diffusion process, and 2) the difficulty in identifying the essential words for generating the desired image. In fact, as described in Fig. 2, a sequence of the hard prompts contain meaningless as well as useful words.

An aim of this paper is to propose a novel prompt generation technique that generates fully interpretable prompts, thereby reducing the need for trial-and-error in text-to-image generation. Essence of our technique, henceforth referred to as visually guided decoding (VGD), is to generate contextually meaningful sentence for hard prompt through the token generation process of large language models (LLMs) (Ouyang et al., 2022; Touvron et al., 2023). In order to align the generated token in the provided images, we use the CLIP model in the token generation process (see Fig. 3 (b)). This process ensures that the resulting prompt is not only semantically relevant to the image but also easily understood by humans. The major benefit of VGD is that one can fully understand the meaning of the prompt (*interpretability*) and synthesize images for various applications (*generalizability*) (*e.g.*, style transfer, image editing, and image variation). Also, when users want to modify objects, background, and/or style, one can easily change the generated prompt (*flexibility*).

In our experiments, we show that VGD qualitatively and quantitatively achieves state-of-the-art (SOTA) performance, demonstrating superior interpretability, generalizability, and flexibility in text-to-image generation compared to previous soft and hard prompt inversion methods. We also show that VGD is compatible with a combination of various LLMs (*i.e.*, LLaMA2, LLaMA3, Mistral) and

image generation models (*i.e.*, DALL-E 2, MidJourney, Stable Diffusion 2). Since VGD seamlessly stitches LLMs and text-to-image models without additional training (*i.e.*, gradient-free), VGD provides a flexible and efficient solution for chat-interfaced image generation services using LLM (*e.g.*, DALL-E extension on ChatGPT) (see Fig. 1).

## 2 BACKGROUND AND RELATED WORKS

### 2.1 CLIP MODEL

CLIP model aligns semantically related visual and textual content within a shared representation space (Radford et al., 2021; Lee et al., 2025). It consists of an image encoder $\text{CLIP}_{\text{img}}(\cdot)$ and a text encoder $\text{CLIP}_{\text{txt}}(\cdot)$. The image encoder encodes an input image $I$ into a visual embedding $\mathbf{f}_{\text{img}} = \text{CLIP}_{\text{img}}(I)$. The text encoder processes the input text $T = [x_1^{\text{txt}}, x_2^{\text{txt}}, ..., x_N^{\text{txt}}]$ to generate a textual embedding $\mathbf{f}_{\text{txt}} = \text{CLIP}_{\text{txt}}(T)$. CLIP is trained using a contrastive learning approach such that the similarity between $\mathbf{f}_{\text{txt}}$ and $\mathbf{f}_{\text{img}}$ for matched sentence-image pair is maximized, while the similarity for unmatched pairs is minimized. The similarity is defined as:

$$\text{Sim}(\mathbf{f}_{\text{txt}}, \mathbf{f}_{\text{img}}) \coloneqq s \frac{\mathbf{f}_{\text{txt}} \cdot \mathbf{f}_{\text{img}}}{\|\mathbf{f}_{\text{txt}}\|_2 \|\mathbf{f}_{\text{img}}\|_2} \tag{1}$$

where $s$ is a scalar scaling factor. Then, the training objective of CLIP is formulated as:

$$\underset{\text{CLIP}_{\text{img}}, \text{CLIP}_{\text{txt}}}{\textbf{Maximize}} \quad \log P\left(T|I\right) + \log P\left(I|T\right), \tag{2}$$

$$P(T|I) = \frac{\exp\left(\text{Sim}(\mathbf{f}_{\text{txt}} \cdot \mathbf{f}_{\text{img}})\right)}{\sum_{j=1}^{B} \exp\left(\text{Sim}(\mathbf{f}_{\text{txt}_j} \cdot \mathbf{f}_{\text{img}})\right)}, \quad P(I|T) = \frac{\exp\left(\text{Sim}(\mathbf{f}_{\text{txt}} \cdot \mathbf{f}_{\text{img}})\right)}{\sum_{j=1}^{B} \exp\left(\text{Sim}(\mathbf{f}_{\text{txt}} \cdot \mathbf{f}_{\text{img}_j})\right)}, \tag{3}$$

where $B$ indicates the number of text-image pairs in the mini-batch.

### 2.2 TEXT-TO-IMAGE DIFFUSION MODEL

Text-to-image models generate an image $I$ that maximizes $P(I|T)$ given textual description $T$. Modern text-to-image models, such as Stable Diffusion, are based on Latent Diffusion Model (LDM), which usually takes CLIP text embedding $\mathbf{f}_{\text{txt}}$ as a condition to generate an image. Our goal is to find the optimal text condition $T$ that yields the image containing desired visual concepts.

### 2.3 PROMPT INVERSION

**Soft Prompt Inversion** Following prompt-tuning approach Li & Liang (2021), Soft Prompt Inversion techniques (Gal et al., 2023; Kumari et al., 2023) extend the vocabulary of the model with a new special token $S_*$, which embeds the objects from the provided images. During the generation process, the continuous embedding vector of $S_*$ is prepended to the embeddings of the input text tokens (*e.g.*, "A photo of $S_*$", "A rendition of $S_*$") and then used as an input to the LDM model. The soft prompt is a high-dimensional vector with real elements so that it is non-human-readable.

**Hard Prompt Inversion** To mitigate the limitations of Soft Prompt Inversion, hard prompt techniques has been proposed (Wen et al., 2024; Mahajan et al., 2024). In this scheme, the generated soft prompt is mapped to the nearest word token to produce the hard prompt. Often, the sequence of hard prompts consists of unrelated words, so that it is very difficult to identify words to be modified in the desired image generation. Moreover, hard prompt inversion involves a complex training process due to the discrete optimization and multi-stage pipelining.

**Image Captioning** One might think of an idea to generate an image directly from captions using LMM (Li et al., 2022; 2023; Alayrac et al., 2022; Liu et al., 2024). This approach is conceptually simple but the generated captions lack the fine details necessary for detailed image synthesis control. To complement the missing information of image captions, prompt generation services like CLIP-Interrogator [1] have been introduced (see Appendix A.4 for more details).

---

[1] https://github.com/pharmapsychotic/clip-interrogator

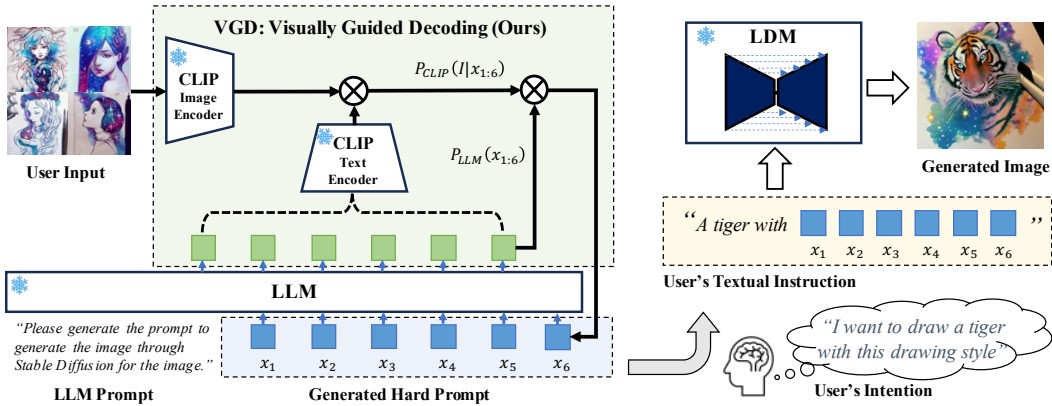

Figure 4: Overview of the proposed hard prompt inversion method, VGD, which seamlessly integrates LLM, CLIP, and LDM for user-friendly image generation process.

## 3 GRADIENT FREE PROMPT INVERSION WITH LANGUAGE MODELS

### 3.1 INTERPRETABILITY DEGRADATION OF PREVIOUS HARD PROMPT INVERSION

The goal of hard prompt inversion is to determine the text prompt $T = [x_1^{\text{txt}}, x_2^{\text{txt}}, ..., x_N^{\text{txt}}]$ that maximizes the LDM's probability $P(I|T)$ for a target image $I$, with each $x_i^{\text{txt}}$ being chosen from the model's vocabulary. By applying Bayes' theorem, this objective can be reformulated as

$$P(I|T) = \frac{P(I)P(T|I)}{P(T)}. \tag{4}$$

Since the prior probability of the image $P(I)$ is independent of $T$, finding the optimal $\hat{T}$ is equivalent to finding $T$ that maximizes $P(T|I)/P(T)$. That is,

$$\hat{T} = \underset{T}{\text{argmax}} \frac{P(T|I)}{P(T)}. \tag{5}$$

As indicated in Eq. 5, the inversion process maximizes $P(T|I)$ (*a.k.a.,* image captioning objective) while simultaneously minimizing $P(T)$, the prior probability of text. In the $P(T)$ minimization process, the hard prompt $T$ might contain uncommon or awkward phrasing, resulting in a degradation of interpretability. We argue that this is why conventional hard prompt inversion techniques exhibit lower interpretability (see Section 4.3 for qualitative comparison).

### 3.2 VISUALLY GUIDED DECODING

**Step 1 - Problem Formulation** Our goal is to find an *interpretable* text prompt without training prompt embeddings, using the *gradient-free* approach. Inspired by the noisy channel model (Jurafsky, 2000; Brown et al., 1993), widely used in machine translation and speech recognition, we model the prompt discovery process by integrating an LDM objective with a regularization term for language modeling. The objective is formulated as

$$\hat{T} = \underset{T}{\text{argmax}} \, P(I|T)P(T). \tag{6}$$

**Step 2 - Approximation with CLIP Score** Computing $P(I|T)$ using forward and reverse diffusion processes in every token generation step is computationally intractable. To alleviate this computational burden, we approximate $P(I|T)$ using CLIP, such that $P(I|T) \approx P_{\text{CLIP}}(I|T)$. This approximation is justified by the fact that Stable Diffusion utilizes the frozen CLIP text encoder. Empirically, we demonstrate that when a CLIP model different from the one aligned with Stable Diffusion is used, the approximation no longer holds, leading to a degradation in performance (see Section 4.4 for the ablation study on CLIP image encoders).

For the prior probability of text $P(T)$, we leverage an external language model (e.g., LLaMA, Mistral, GPT). Our main objective is then formulated as

$$\hat{T} = \underset{T}{\operatorname{argmax}}\, P_{\text{CLIP}}\left(I|T\right)P_{\text{LLM}}\left(T\right)^{\alpha}, \tag{7}$$

where $\alpha$ is a hyperparameter.

**Step 3 - Token-by-Token Generation**   We can express Eq. 7 using a left-to-right decomposition of the text probability $P_{\text{LLM}}(T)$. That is,

$$P_{\text{CLIP}}\left(I|x_{1:N}^{\text{txt}}\right) \prod_{i=1}^{N} P_{\text{LLM}}\left(x_i^{\text{txt}}|x_{1:i-1}^{\text{txt}}\right)^{\alpha}. \tag{8}$$

This decomposition facilitates token-by-token text generation using a beam search decoding strategy (Anderson et al., 2017; Post & Vilar, 2018; Hu et al., 2019; Holtzman et al., 2020). Specifically, we iteratively select the $i$-th token $x_i^{\text{txt}}$ that maximizes the objective in Eq. 8. In doing so, VGD generates prompts that are semantically aligned with the target image (*via* CLIP) while ensuring linguistic coherence and interpretability (*via* LLM).

### 3.3 Implementation of Visually Guided Decoding

**LLM Prompting**   Instead of fine-tuning LLM for each image generation task, we design specific system and user prompts tailored to different tasks. Using the designed system and user prompt, LLM is queried to generate tokens as if it were creating text prompts for text-to-image models (see Appendix A.1 for details).

**Beam Initialization**   Providing LLM with image-related words at the start of the generation process increases the probability of generating tokens that align with the image in subsequent steps. To do so, we first compare all text embeddings $\mathbf{f}_{\text{txt}}$ in CLIP vocabulary with the target image embedding $\mathbf{f}_{\text{img}}$, selecting the top-$M$ tokens $x_{1:M}^{\text{txt}}$ as the initial input prefix for LLM. Empirically, we find that even $M = 1$ is sufficient for VGD. This process is computationally efficient since all text embeddings can be precomputed.

**Beam Expansion and Pruning**   Since CLIP does not provide next-token probabilities, we use LLM's next-token predictions to select beam search candidates. Specifically, for a beam width of $K$, each beam is expanded by appending the $K$ highest-probability next-token candidates based on $P_{\text{LLM}}(x_i^{\text{txt}}|x_{1:i-1}^{\text{txt}})$. The expanded set of $K^2$ candidates is then pruned to retain only $K$ beams according to Eq. 8. Unless otherwise specified, we set $K = 10$ for the experiments.

**Beam Search Termination**   The beam search terminates when either 1) beam expansion fails to improve the score (Eq. 8), or 2) the predefined maximum prompt length is reached. The final result $\hat{T}$ is then selected from the remaining candidates based on the highest score.

## 4 Experiments

### 4.1 Setup

**Datasets**   We conduct experiments on four datasets with diverse distributions: LAION-400M (Schuhmann et al., 2021; 2022), MS COCO (Lin et al., 2014), Celeb-A (Liu et al., 2015), and Lexica.art [2]. Following PEZ (Wen et al., 2024), we randomly sample 100 images from each dataset and evaluate prompt inversion methods across 5 runs using different random seeds. See Appendix A.3 for more details.

**Baselines**   We compare the proposed method with the hard prompt inversion (PEZ (Wen et al., 2024)), soft prompt inversion (Textual Inversion (Gal et al., 2023)), LLaVA-1.5 (Liu et al., 2024) generated caption, and LLaVA-1.5 combined with CLIP-Interrogator. Images are generated with the Stable Diffusion 2.1-768 model across all comparisons (Podell et al., 2024).

---

[2]https://huggingface.co/datasets/Gustavosta/Stable-Diffusion-Prompts

Table 1: Image quality (CLIP-I score) and prompt quality (BERTScore) comparison.

| Method | #Tokens | LAION | MS COCO | Celeb-A | Lexica.art | MS COCO (BERTScore) | | | Lexica.art (BERTScore) | | |
|---|---|---|---|---|---|---|---|---|---|---|---|
| | | CLIP-I Score | | | | Precision | Recall | F1 | Precision | Recall | F1 |
| Textual Inversion | 1 | 0.388 | 0.569 | 0.522 | 0.697 | - | - | - | - | - | - |
| LLaVA-1.5 | 32 | 0.513 | 0.642 | 0.463 | 0.580 | 0.914 | 0.918 | 0.916 | 0.858 | 0.780 | 0.817 |
| + CLIP Interrogator | ~77 | 0.540 | 0.685 | 0.511 | 0.762 | 0.794 | 0.898 | 0.843 | 0.818 | 0.819 | 0.819 |
| LLaVA-1.5 + **VGD** | 32 | 0.573 | 0.718 | 0.546 | 0.769 | 0.831 | 0.879 | 0.854 | 0.835 | 0.793 | 0.813 |
| | ~77 | 0.569 | 0.724 | 0.544 | 0.785 | 0.800 | 0.874 | 0.835 | 0.810 | 0.794 | 0.802 |
| PH2P | 16 | 0.442 | 0.589 | 0.413 | 0.678 | 0.800 | 0.842 | 0.820 | 0.805 | 0.780 | 0.792 |
| PEZ | 16 | 0.538 | 0.687 | 0.622 | 0.743 | 0.760 | 0.834 | 0.795 | 0.772 | 0.783 | 0.777 |
| | 32 | 0.530 | 0.685 | 0.619 | 0.745 | 0.736 | 0.830 | 0.780 | 0.752 | 0.784 | 0.768 |
| | 64 | 0.507 | 0.670 | 0.584 | 0.728 | 0.715 | 0.825 | 0.766 | 0.734 | 0.783 | 0.758 |
| **VGD (Ours)** | 16 | 0.484 | 0.650 | 0.482 | 0.700 | 0.833 | 0.862 | 0.847 | 0.827 | 0.779 | 0.802 |
| | 32 | 0.493 | 0.670 | 0.506 | 0.735 | 0.818 | 0.868 | 0.842 | 0.816 | 0.786 | 0.801 |
| | 64 | 0.511 | 0.678 | 0.514 | 0.753 | 0.787 | 0.863 | 0.823 | 0.799 | 0.789 | 0.794 |
| | ~77 | 0.510 | 0.678 | 0.513 | 0.754 | 0.788 | 0.864 | 0.824 | 0.801 | 0.791 | 0.795 |

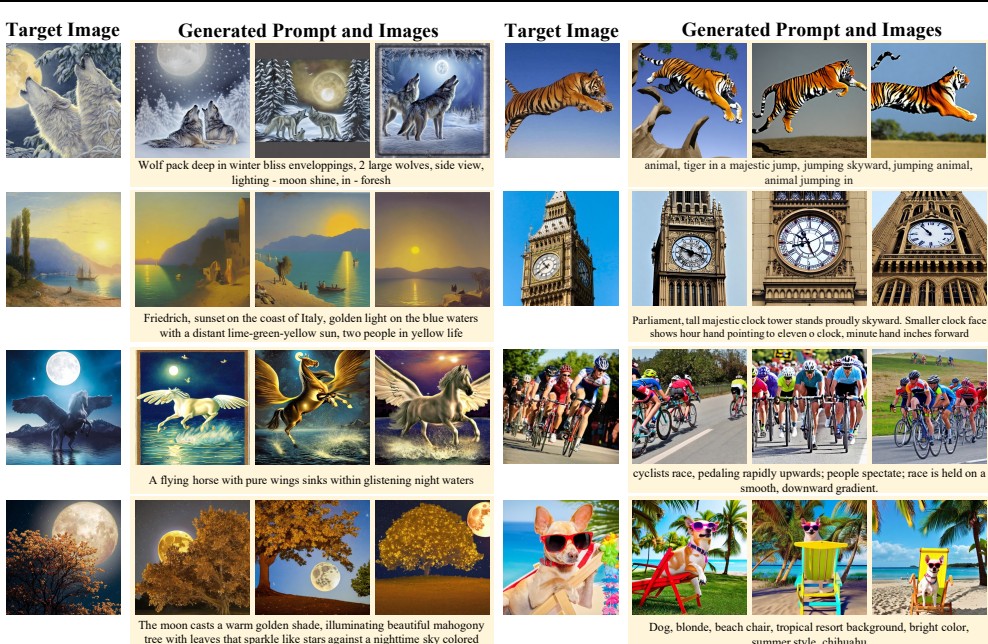

Figure 5: Generated images using hard prompts produced by the proposed VGD.

**Evaluation Metric** We evaluate the quality of the prompt via image embedding similarity between the target image and an image generated using the generated hard prompt. We call this similarity-based metric CLIP-I score. To ensure fairness, we utilize different CLIP models for generation and evaluation: CLIP-ViT-H-14 for generation and larger CLIP-ViT-G-14 for similarity evaluation. Following (Mahajan et al., 2024), we also compare the contextual similarity between the generated prompt and the ground-truth annotations (captions) of the target image, using BERTScore (Zhang et al., 2020) as metric.

## 4.2 IMAGE GENERATION WITH VGD PROMPTS

In Fig. 5, we qualitatively demonstrate that VGD generates realistic and diverse images without overfitting to the original target image.

**Qualitative Evaluation** As shown in Table 1, the images generated by VGD are the most similar to the original images, as measured by the CLIP-I score, even without using a gradient-based optimization process like PEZ. When using LLaVA-1.5 as the language model, VGD achieves SOTA CLIP-I scores across all four datasets, surpassing the strong baseline that exploits an external database (LLaVA 1.5 + CLIP Interrogator) by a considerable margin. Furthermore, unlike PEZ, VGD exhibits a positive correlation between the CLIP-I score and the number of tokens, highlighting the effectiveness of VGD's token-by-token generation process.

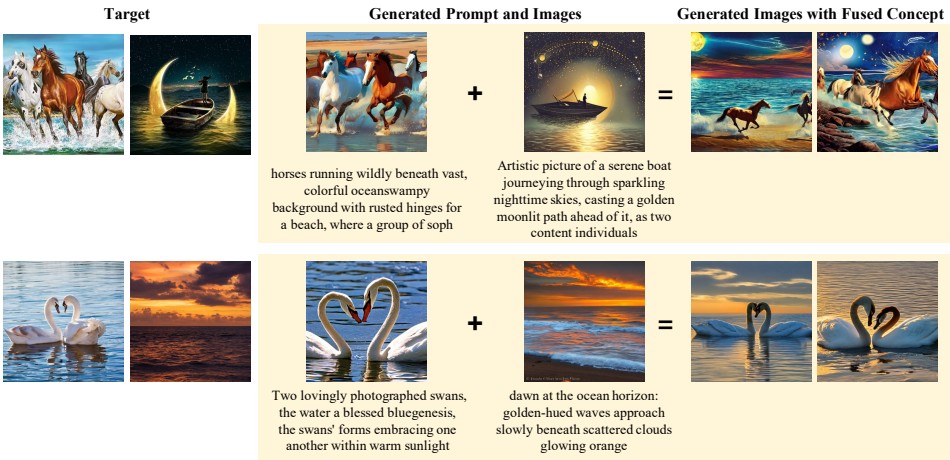

Figure 6: Multi-concept image generation with VGD. We show that the hard prompts obtained from two different images can be concatenated to fuse the semantic concepts.

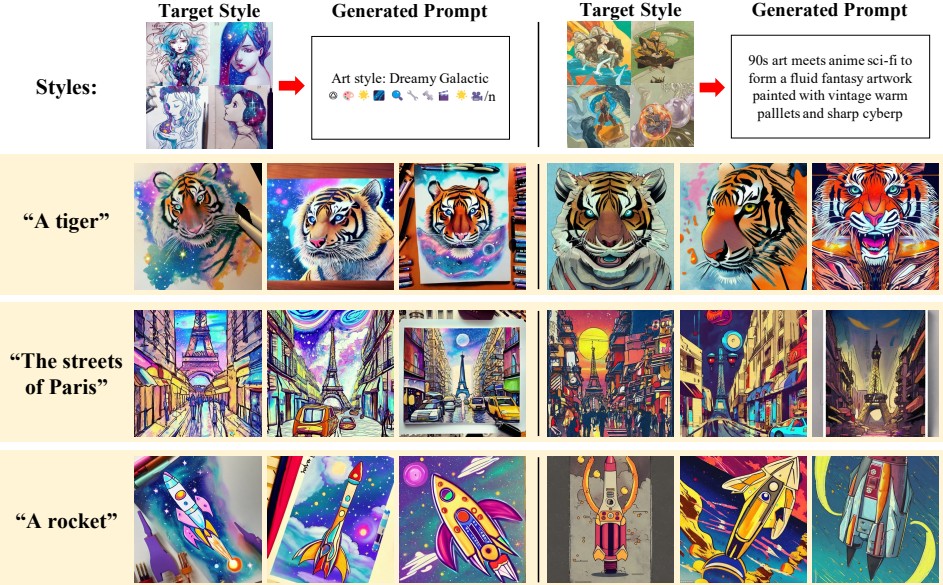

Figure 7: Decoded hard prompt for style transfer. Given several sample images with the same style, we can extract the style with a hard prompt and transfer it to other objects or scenes.

**Multi-concept Generation** VGD is capable of fusing multiple concepts from different images without additional effort. For each image with its own distinct concept, we individually extract prompts and concatenate them to generate a single image that combines their concepts. Fig. 6 presents two examples of multi-concept generation, illustrating that VGD accurately extracts the key elements from each image so that the generated image can contain multiple concepts without omitting any. We show the example of multi-concept generation of 5 images with VGD and PEZ in Fig. 21 of the Appendix.

**Style Transfer** VGD can also be easily adapted to style transfer. Given several examples (typically 4 images) that share the same style, we extract their shared style characteristics as a single hard prompt and use this prompt to apply the style to new objects or scenes. Fig. 7 shows two examples of style transfer, demonstrating that VGD correctly embeds the shared style elements in the prompt, which can be easily combined with new objects.

### 4.3 QUALITY OF VGD PROMPTS

**Quantitative Evaluation**  To assess the coherence of generated prompts, we measure the BERTScore between the prompts generated by each method and the ground-truth captions. As shown in Table 1, VGD achieves a significant improvement in BERTScore compared to PEZ, especially when using more than 64 tokens. Although LLaVA-based baselines show higher BERTScore, their CLIP-I scores are worse than those of VGD. This implies that the LLaVA captions fail to describe the target image in detail, even though the text itself may be semantically similar to ground-truth captions.

**Interpretability**  When a prompt closely aligns with the image content and is interpretable by humans, it becomes easier for users to modify the image in the desired direction by making slight edits to the prompt. To evaluate the interpretability of the generated prompts, we make minimal modifications to the prompt and observe the effects. For example, we change "winter" to "spring" and see if the generated image successfully reflects the change. As illustrated in Fig. 9, we compare VGD with previous gradient-based prompt inversion techniques, namely PEZ (hard) and Textual Inversion (soft).

The results indicate that VGD exhibits superior interpretability. When we adjust the number of objects, background elements, atmosphere (seasons), and species in the prompt, VGD successfully incorporates these changes. The relevant terms are already present in the original prompt, enabling accurate image generation with the modified prompts. In contrast, the competitors struggle to produce the desired outcomes. For instance, when we try to generate four wolves, both PEZ and Textual Inversion fail to change the number of wolves included.

**Prompt Distillation**  We can also utilize prompt inversion to reduce the length of prompts while preserving their capability, *a.k.a.* prompt distillation. Long prompts may contain redundant and unimportant information, especially when hand-crafted. This becomes problematic, particularly when there exists a limit on the number of tokens that the model can process. Using VGD, we can generate a shorter prompt that preserves the key aspects of the original prompt. As illustrated in Fig. 10, the images generated with distilled prompts are very similar to the original image. In addition, images generated with distilled prompts do not show significant performance degradation on CLIP-I score compared to the original prompt, as shown in Fig. 8.

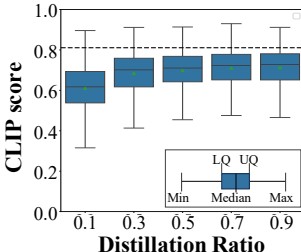

Figure 8: Prompt distillation on Lexica.art dataset.

### 4.4 EFFECT OF CLIP AND LANGUAGE MODEL

Table 2: Comparison of VGD variants.

| Model | CLIP Score | BERTScore | | |
|---|---|---|---|---|
| | | Precision | Recall | F1 |
| **VGD** | 0.735 | 0.816 | 0.786 | 0.801 |
| LLM only | 0.487 | 0.811 | 0.773 | 0.791 |
| CLIP only | 0.768 | 0.727 | 0.779 | 0.751 |

Table 3: Ablation on CLIP models.

| CLIP Model | #Tokens | CLIP Score |
|---|---|---|
| **laion/ViT-H-14 (original)** | 32 | 0.735 |
| openai/ViT-B-32 | 32 | 0.655 |
| openai/ViT-L-14 | 32 | 0.626 |

**Role of CLIP and LLM**  To investigate the effect of using both CLIP and LLM, we modify VGD in two ways and compare these variants on the Lexica.art. In the 'LLM only' variant, VGD selects the next token based solely on $P_{\text{LLM}}(x_i^{\text{txt}}|x_{1:i-1}^{\text{txt}})$ without considering CLIP score, operating like typical LLM text decoding. In the 'CLIP only' variant, VGD determines the next token using only the CLIP model. Table 2 shows that the 'LLM only' variant suffers from a drastic performance degradation in the CLIP score because it does not consider the target image at all. On the other hand, the 'CLIP only' variant achieves the highest CLIP score but the lowest BERTScore, suggesting that the generated prompt is less coherent and harder to interpret. In comparison, the proposed VGD demonstrates strong and balanced performance on the CLIP score and BERTScore.

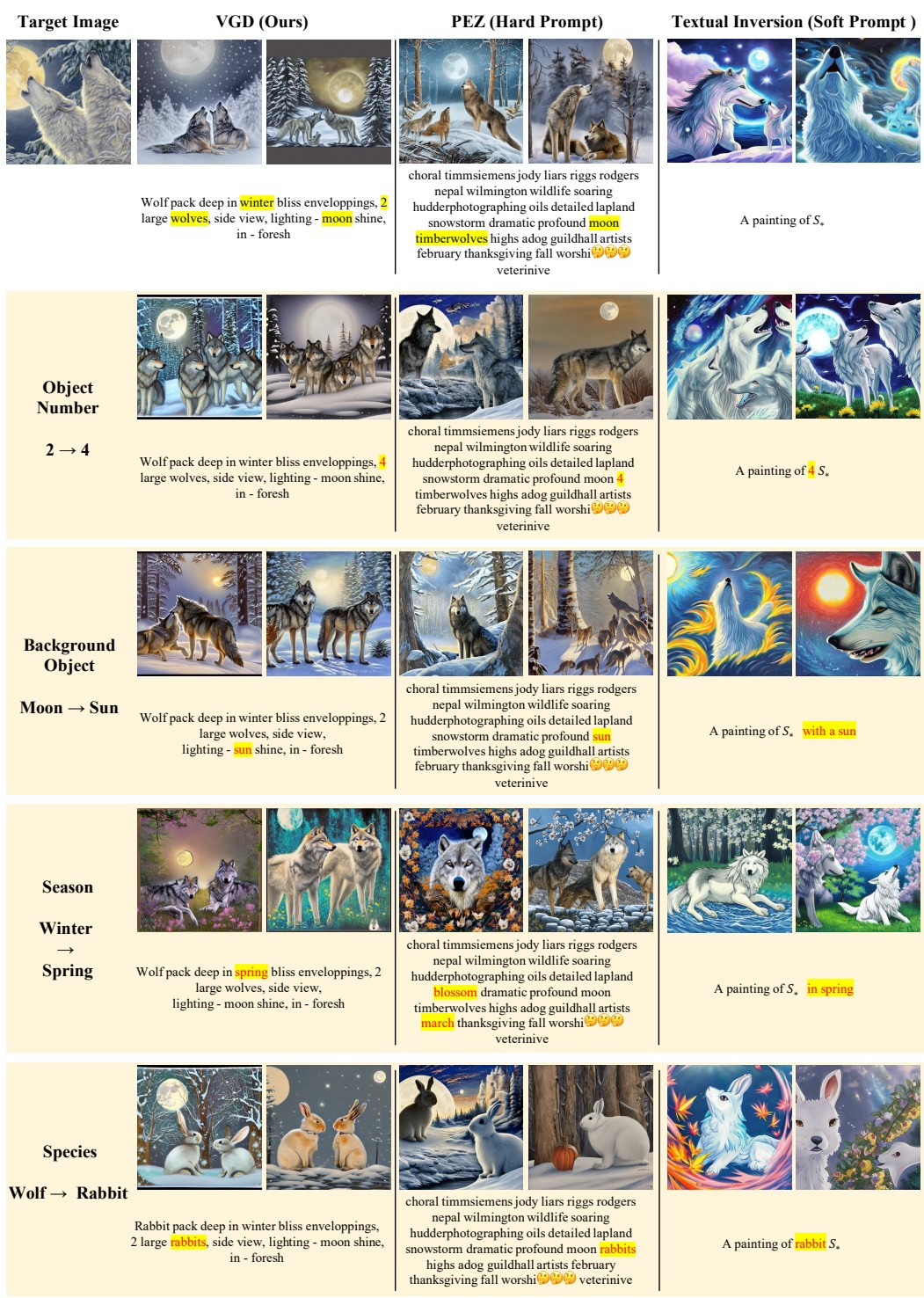

Figure 9: Prompt interpretability. With the good interpretability of the prompt, we can modify them to edit the original image to the desired direction.

**CLIP Model Ablation**   VGD leverages the same CLIP model that is used in the text-to-image model, Stable Diffusion 2.1. To assess the importance of this alignment, we perform an ablation study using two alternative CLIP models: `openai/ViT-B-32` and `openai/ViT-L-14`. Table 3 shows that using a misaligned CLIP model leads to significant performance degradation. This

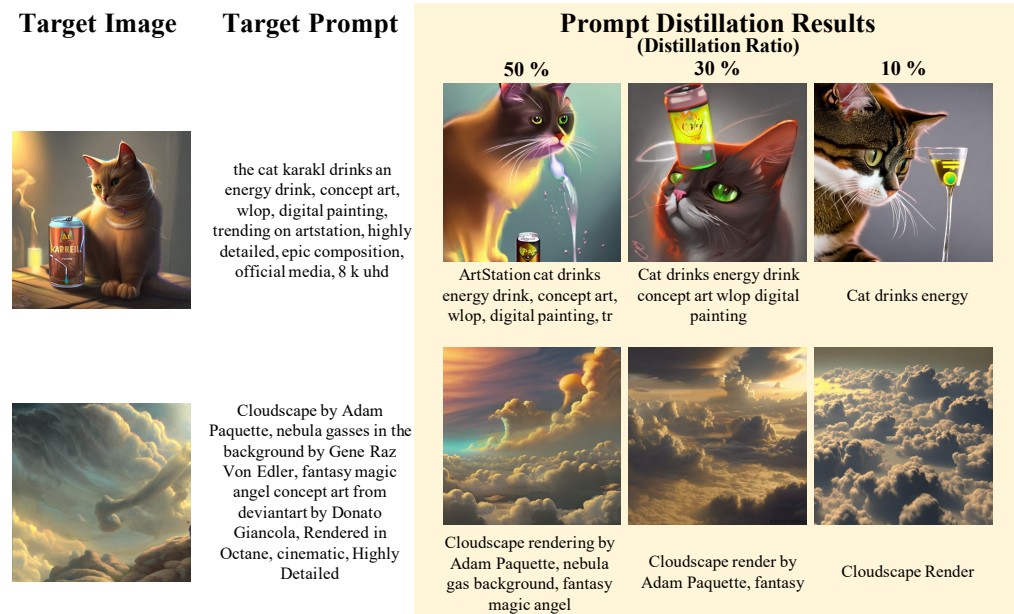

Figure 10: Prompt distillation results generated by the proposed VGD. With VGD, we can distillate the hard prompt that can still generate images similar in concept to the original.

Table 4: Ablation on language models.

| LLM | #Tokens | LAION | MS COCO | Celeb-A | Lexica.art | MS COCO | | | Lexica.art | | |
|-----|---------|-------|---------|---------|------------|---------|--------|-----|------------|--------|-----|
| | | CLIP-I Score | | | | Precision | Recall | F1 | Precision | Recall | F1 |
| Mistral-7B | 16 | 0.443 | 0.615 | 0.422 | 0.706 | 0.808 | 0.850 | 0.828 | 0.815 | 0.780 | 0.797 |
| | 32 | 0.463 | 0.656 | 0.466 | 0.729 | 0.803 | 0.858 | 0.829 | 0.808 | 0.787 | 0.797 |
| | 64 | 0.497 | 0.680 | 0.476 | 0.751 | 0.783 | 0.858 | 0.818 | 0.791 | 0.789 | 0.790 |
| LLaMA2-7B | 16 | 0.484 | 0.650 | 0.482 | 0.700 | 0.833 | 0.862 | 0.847 | 0.827 | 0.779 | 0.802 |
| | 32 | 0.493 | 0.670 | 0.506 | 0.735 | 0.818 | 0.868 | 0.842 | 0.816 | 0.786 | 0.801 |
| | 64 | 0.511 | 0.678 | 0.514 | 0.753 | 0.787 | 0.863 | 0.823 | 0.799 | 0.789 | 0.794 |
| LLaMA3-8B | 16 | 0.480 | 0.648 | 0.521 | 0.731 | 0.820 | 0.861 | 0.840 | 0.829 | 0.784 | 0.806 |
| | 32 | 0.506 | 0.661 | 0.540 | 0.752 | 0.805 | 0.864 | 0.833 | 0.817 | 0.790 | 0.803 |
| | 64 | 0.515 | 0.685 | 0.543 | 0.769 | 0.769 | 0.858 | 0.811 | 0.796 | 0.794 | 0.795 |

highlights the importance of proper approximation $P(T|I) \simeq P_{\text{CLIP}}(T|I)$. When a different CLIP model is used, the approximation no longer holds, leading to a drop in performance.

**Language Model Ablation**  We conduct an ablation study using three different LLMs: Mistral-7B (Jiang et al., 2023), LLaMA2-7B (Touvron et al., 2023), and LLaMA3-8B (Dubey et al., 2024). As shown in Table 4, although LLaMA3-8B exhibits the best performance, the overall differences among the models are marginal. This indicates that VGD is not dependent on the specific language model used, allowing users to choose an LLM that best suits their system.

## 5 CONCLUSION

This paper tackles the difficulty of creating effective textual prompts for advanced text-to-image models like DALL-E and Stable Diffusion. Traditional prompt inversion methods often lack interpretability and coherence, limiting their usefulness. We introduce VGD, a gradient-free technique that combines LLMs with CLIP-based guidance to generate clear and semantically accurate prompts. VGD enhances prompt interpretability, generalization, and flexibility without requiring additional training. Our experiments show that VGD outperforms existing methods, producing more understandable and contextually relevant prompts. By integrating seamlessly with various LLMs such as LLaMA2, LLaMA3, and Mistral, VGD provides an efficient and versatile solution for improving user interactions with text-to-image generative models.

ACKNOWLEDGMENTS

This work was supported by Institute of Information & communications Technology Planning & Evaluation (IITP) grant funded by the Korea government(MSIT) (IITP-2023-RS-2023-00256081 and RS-2024-00398157. The authors would like to thank Jinwoo Son for his valuable assistance in proofreading this manuscript.

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

# A  DETAILS

## A.1  PROMPT SETUP FOR LARGE LANGUAGE MODELS

We use different system and user prompts for different image generation tasks. Tables 5 and 6 show the prompts used for the experiments.

Table 5: LLM prompt setting for different image generation tasks.

---

**System Prompt**

You are a respectful and honest visual description generator for Stable Diffusion text prompt.
Answer in 1 sentence and do not mention anything other than the prompt. Do not mention 'description'.

---

**User Prompt (Prompt Inversion)**

Please generate the diffusion prompt on the given condition containing the objects, people, background, and the style of the image:

---

**User Prompt (Style Transfer)**

Please generate the diffusion prompt of the image style based on the given condition containing the painting style, color, and shapes of the image:

---

**User Prompt (Prompt Distillation)**

Please generate the diffusion prompt within ${max_length} tokens so that you can generate same images with a given prompt: ${target_prompt}

---

**Model prompt**

Answer: Sure, here is a prompt for stable diffusion within ${model.max_length} tokens:

---

Table 6: LLaVA prompt setting for different image generation tasks.

---

**Prompt (Image Captioning)**

USER: < image > Describe the scene in this image with one sentence.
ASSISTANT:

---

**Prompt (VGD)**

USER: < image > Please generate the diffusion prompt containing the objects, people, background and the style of the image.
ASSISTANT: Sure, here is a prompt for stable diffusion within ${model.max_length} tokens:

---

## A.2  HYPERPARAMETERS AND MODELS

The beam width $K$ is set to 10. The balancing hyperparameter $\alpha$ is set to 0.67. For VGD generation, we used `laion/CLIP-ViT-H-14-laion2B-s32B-b79K`[3]. For CLIP-I score evaluation, we used `laion/CLIP-ViT-g-14-laion2B-s12B-b42K`[4]. We used Stable Diffusion `stabilityai/stable-diffusion-2-1`[5] for text-to-image model.

## A.3  DATASET CHARACTERISTICS

LAION-400M (Schuhmann et al., 2021; 2022) comprises 400 million diverse CLIP-filtered images. MS COCO (Lin et al., 2014) mainly contains real-life photographs with multiple common objects,

---

[3]https://huggingface.co/laion/CLIP-ViT-H-14-laion2B-s32B-b79K
[4]https://huggingface.co/laion/CLIP-ViT-g-14-laion2B-s12B-b42K
[5]https://huggingface.co/stabilityai/stable-diffusion-2-1

whereas Celeb-A (Liu et al., 2015) consists of celebrity portraits. Lexica.art contains AI-generated paintings with their prompts. While different datasets exhibit different characteristics, VGD works effectively regardless of the data type.

### A.4 CLIP-INTERROGATOR'S APPROACH

CLIP-Interrogator supplements missing details in generated image captions by extensive trial-and-error approach. Specifically, CLIP-Interrogator augments the caption with additional tokens selected from a pre-collected bank of 100K keywords (*e.g.*, artist names, styles, mediums, movements), and compare each with the target image through CLIP model to filter out image-irrelevant keywords. While effective in generating the hard prompt, CLIP-Interrogator has several drawbacks: 1) an inability to generate images for styles, artists, or objects that are not registered in the keyword bank, and 2) the computational burden of processing 100K phrases through a CLIP encoder. These limitations highlight the challenges of relying solely on image captions for image synthesis, and underscore the need for efficient and flexible hard prompt generation techniques.

## B ADDITIONAL RELATED WORKS

### B.1 PROMPT EDITING FOR TEXT-TO-IMAGE MODELS

To fully harness the capabilities of text-to-image models, various approaches have been proposed to tailor text prompts to the user's specific intentions (Hertz et al., 2023; Brooks et al., 2023; Wang et al., 2024) Notably, a rich text editor has been introduced, enabling users to design complex textual guidance to image generation models (Ge et al., 2023). Similarly, Prompt Highlighter, an interactive text-to-image interface utilizing multi-modal LLMs, has been proposed (Zhang et al., 2024). These studies underscore the necessity for user-friendly and controllable text editing methods. Our VGD can be seamlessly integrated into LLM-based user interfaces, as illustrated in Fig. 1. VGD leverages the same token generation process as LLMs, benefiting from the highly optimized inference process without requiring additional training.

### B.2 INTERPRETABILITY OF TEXT-TO-IMAGE MODELS

As text-to-image applications become standard in the industry, interpretability studies for these models have gained significant attention. For instance, research has focused on the cross-attention mechanism to identify the model's attention for each word (Tang et al., 2023). In advance, the diffusion objective has been investigated to understand the internal behavior of text-to-image models (Kong et al., 2024). Building on these findings, recent works modified the model's architecture to more faithfully reflect user's instructions (Orgad et al., 2023; Guo & Lin, 2024). We emphasize that VGD aims to enhance the human interpretability without architectural modifications. We believe that previous works can also be also applied to VGD to further improve usability.

## C ADDITIONAL EXPERIMENTAL RESULTS

### C.1 GENERALIZATION OF GENERATED PROMPTS TO DIFFERENT T2I MODELS

We demonstrate how prompts generated by our method and other baseline methods behave when applied to different text-to-image models (see Fig. 13, 14, 15, and 16). In this experiment, we did not generate any new prompt for the newly tested T2I model (*i.e.*, MidJourney, DALL-E 2), meaning that we simply reused the prompts generated for SD in evaluating the effectiveness of producing high-quality images with various T2I models. In doing so, we can ensure a fair comparison and also highlight the generalizability of the prompts across models. Finally, we would like to point out that this type of generalization cannot be done with soft prompt methods like Textual Inversion, as they are inherently tied to the specific model they are trained on. When compared to hard prompt inversion methods (such as PH2P and PEZ), VGD generates consistently better results in various text-to-image models, which underscores the robustness and flexibility of the proposed VGD.

## C.2   PROMPT GENERATION TIME

We further investigate the efficiency of VGD in comparison with other baseline methods, measured on a single A100 80GB GPU. As shown in Table 7, the proposed VGD needs significantly less time to generate each image; 7x faster than Textual Inversion, 12x faster than PEZ, and more than 2200x faster than PH2P. This is because VGD does not require any training steps and only the decoding process is slightly modified.

Table 7: Prompt generation time of VGD, PEZ, PH2P, Textual Inversion, and CLIP Interrogator.

| Method | PEZ | PH2P | Textual Inversion | CLIP Interrogator | **VGD (ours)** |
|---|---|---|---|---|---|
| Time (sec) | 193.14 | 34264.74 | 103.45 | 24.85 | **15.33** |

## C.3   MORE QUALITATIVE COMPARISON

As shown in Fig. 17, 18 and 19, we conduct a qualitative comparison of image variation and style transfer performance across different methods, including PEZ, Textual Inversion, and CLIP Interrogator. The results demonstrate that, similar to the findings in Table 1, our method outperforms prior works, with CLIP Interrogator coming second.

While CLIP Interrogator captures objects effectively by generating captions using LLaVA (Liu et al., 2024) and predefined rich keyword bank, there are some critical weaknesses in prompts generated by the CLIP Interrogator (see Fig. 17). First, the subsequent keywords extracted from the caption's keyword bank using CLIP similarity often suffer from redundancy and lack of interpretability. Second, CLIP Interrogator tends to generate prompts of full length (77 words), which limits its applicability to diverse tasks like multi-concept image generation and style transfer. Finally, as seen in the prompt example for the tiger image in Fig. 17, the quality of the generated images often decreases when using the extracted keywords, compared to images generated without attaching the keywords. In some cases, this leads to all extracted keywords being rejected entirely.

## C.4   HUMAN EVALUATION

To evaluate whether the images generated by VGD are semantically aligned with the original image and to assess image quality, we conduct human preference evaluation. For this experiment, participants were shown images generated for image variation and style transfer by five different methods (VGD, Textual Inversion, PEZ, PH2P, and CLIP Interrogator) and were asked to select the image most similar to the target image and style (Kumari et al., 2023) (see Fig. 12). The evaluation was conducted using 65 images randomly sampled from Lexica.art dataset. Fig. 11 demonstrates that our method consistently performed best in both image variation and style transfer tasks according to user preferences. Specifically, 60.5% and 52.6% of the participants selected VGD's results as the most similarly generated ones for image variation and most instruction-following generations for style transfer tasks, respectively.

## C.5   BAD CASES

While VGD generally shows decent results, our method sometimes struggles to capture fine-grained details of regional or background objects in complex images. These cases occur when the image contains multiple objects (see Fig. 20). This is because VGD has difficulty in generating prompts for regional or local objects in the background. We believe this phenomenon is mainly due to the CLIP vision encoder's tendency to prioritize main objects in an image (Jing et al.; Paiss et al., 2023; Ranasinghe et al., 2023; Yuksekgonul et al., 2022; Zhong et al., 2022).

## C.6   MULTI-CONCEPT GENERATION

We further explore the potential of VGD by combining five different concepts into a single image. Fig. 21 demonstrates that the prompts generated by VGD can be concatenated to generate a complex image without missing concepts. However, generated prompts from PEZ suffers omits important details, such as the parliamentary, wolves, and the style of The Starry Night. This shows that VGD generates hard prompts that are compact and meaningful.

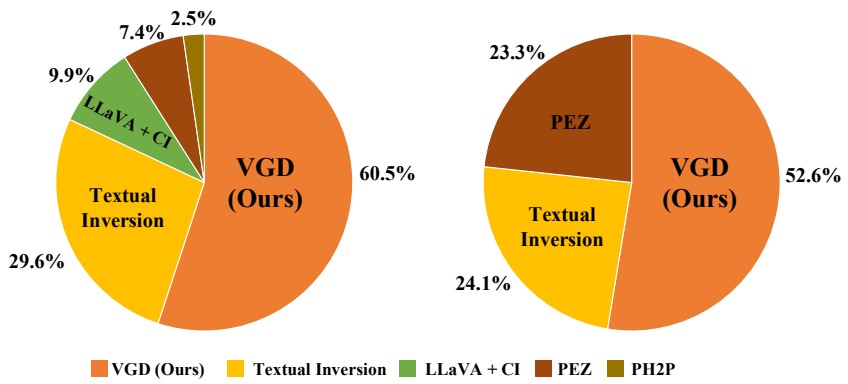

Figure 11: Human preference evaluation for image variation (left) and style transfer (right) tasks using different prompt generation methods. CI indicates CLIP Interrogator. Best viewed in color.

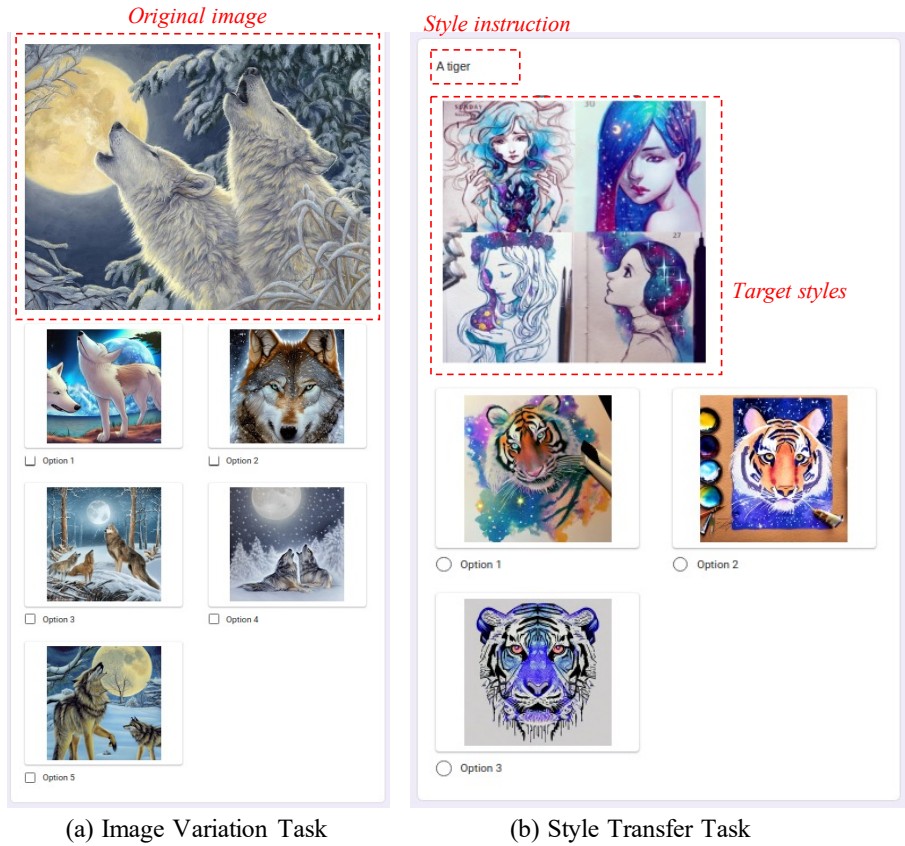

(a) Image Variation Task  (b) Style Transfer Task

Figure 12: Webpage used for human preference evaluation.

## D  LIMITATION AND ETHICAL STATEMENT

### D.1  LIMITATIONS

Although we showcase the generalizability of VGD-generated prompts across multiple text-to-image models, our evaluations primarily focus on popular models such as Stable Diffusion and MidJourney. However, since many other models are trained on similar datasets and follow compara-

Table 8: Prompt interpretability evaluation with Perplexity metric. Lower scores indicate better performance.

| Method | #Tokens | LAION-400M | MS COCO | Celeb-A | Lexica.art |
|---|---|---|---|---|---|
| PEZ | 32 | 813.27 | 766.81 | 971.35 | 659.39 |
| LLaVA-1.5 + CI | ∼77 | 102.62 | 80.97 | 67.59 | 76.04 |
| PH2P | 16 | 1274.14 | 947.29 | 892.04 | 1126.11 |
| **VGD** | 32 | 94.96 | 89.48 | 68.69 | 91.12 |
| **VGD+LLaVA** | 32 | 190.37 | 106.84 | 108.80 | 121.11 |
| **VGD+LLaVA** | 77 | 42.00 | 36.11 | 34.39 | 47.07 |

ble methods and architectures, we anticipate that the results would be consistent across these models as well.

## D.2 ETHICS STATEMENT

The development of VGD raises important ethical considerations, particularly regarding potential misuse. While our method improves the interpretability and generalizability of text-to-image generation, it could also be exploited to generate harmful or inappropriate content, such as deepfakes or offensive imagery. We strongly discourage such uses and emphasize that this method is intended for responsible applications in creative, educational, and professional contexts.

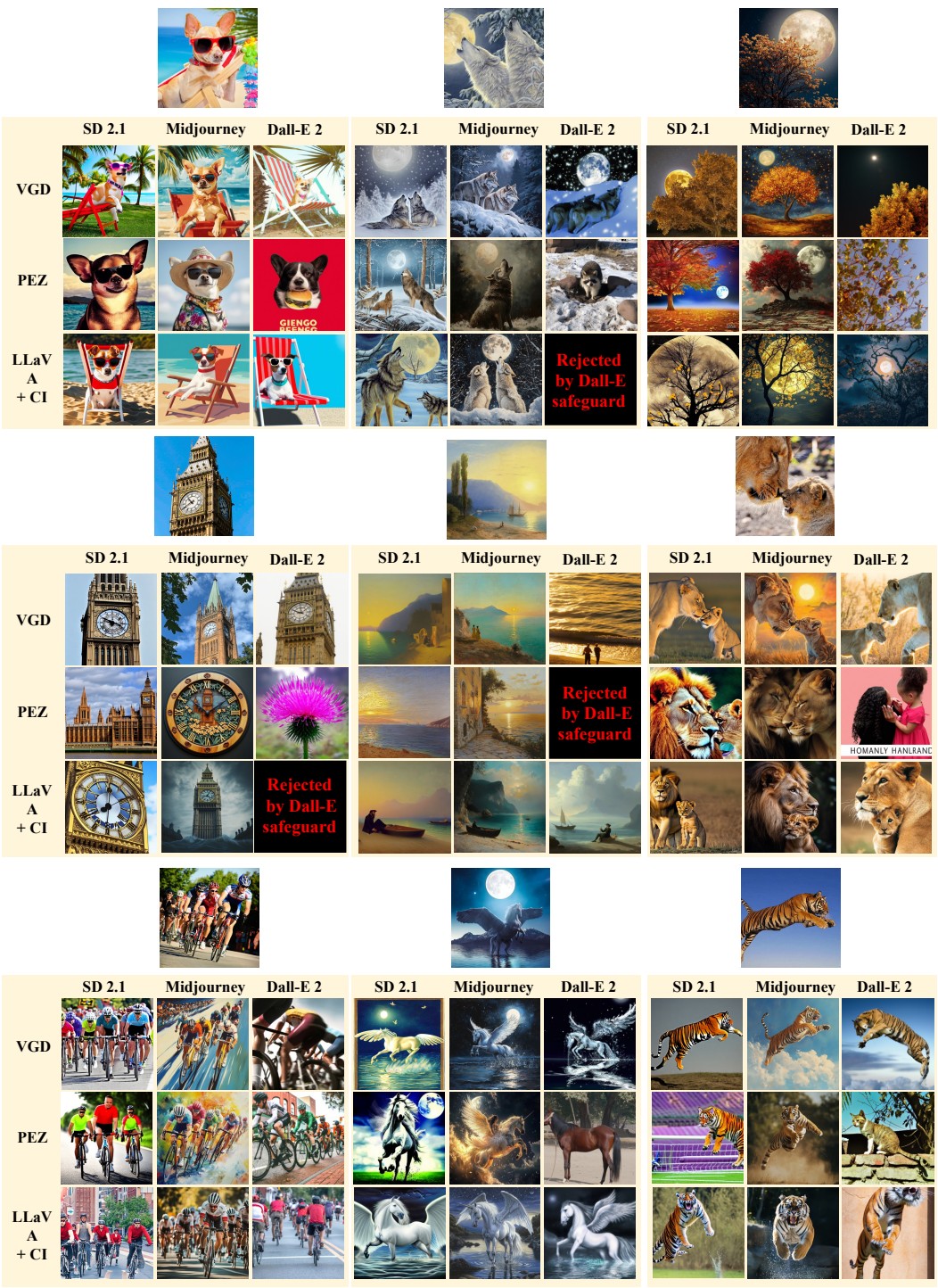

Figure 13: Qualitative examples generated by Stable Diffusion, Midjourney and Dall-E 2 using prompts generated from VGD, PEZ, and LLaVA+CLIP-Interrogator.

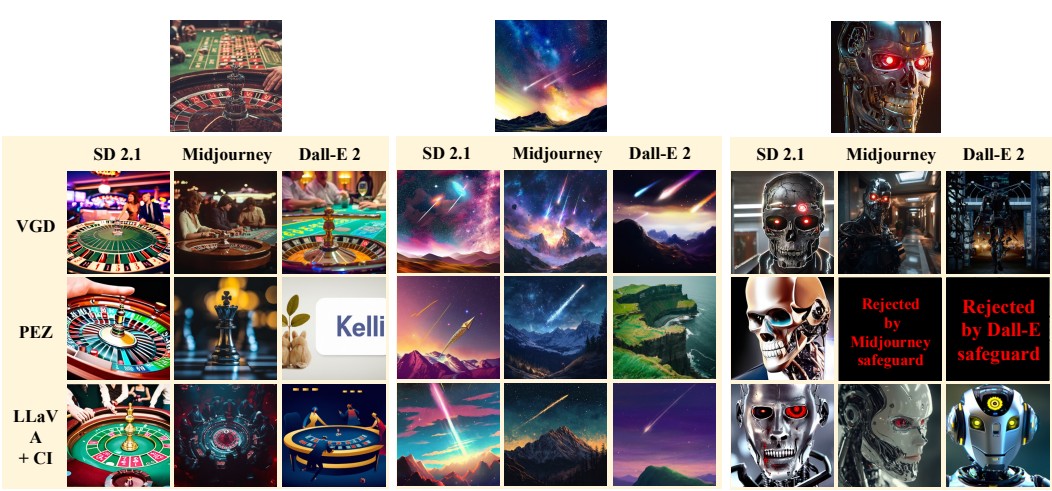

Figure 14: More qualitative examples generated by Stable Diffusion, Midjourney and Dall-E 2 using prompts generated from VGD, PEZ, and LLaVA+CLIP-Interrogator.

Figure 15: More examples generated by MidJourney.

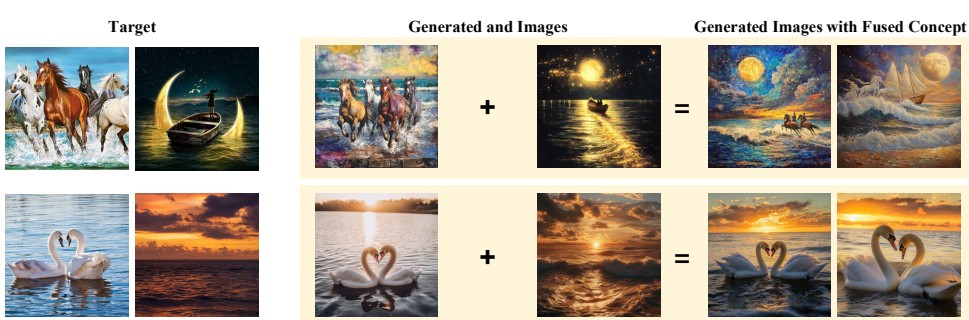

Figure 16: Multi-concept image generation by MidJourney.

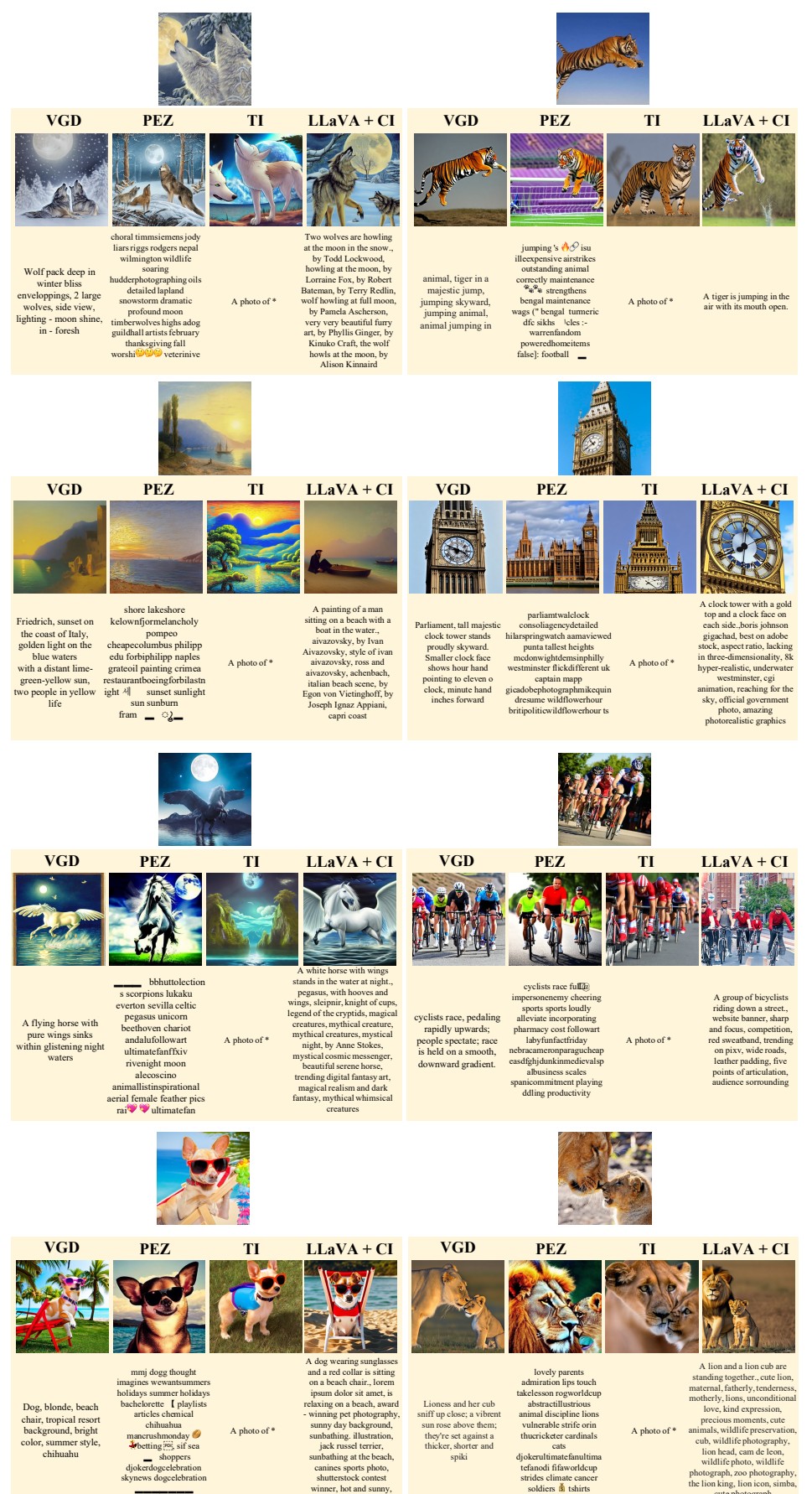

Figure 17: Image variation comparison between VGD and baselines (generated by Stable Diffusion).

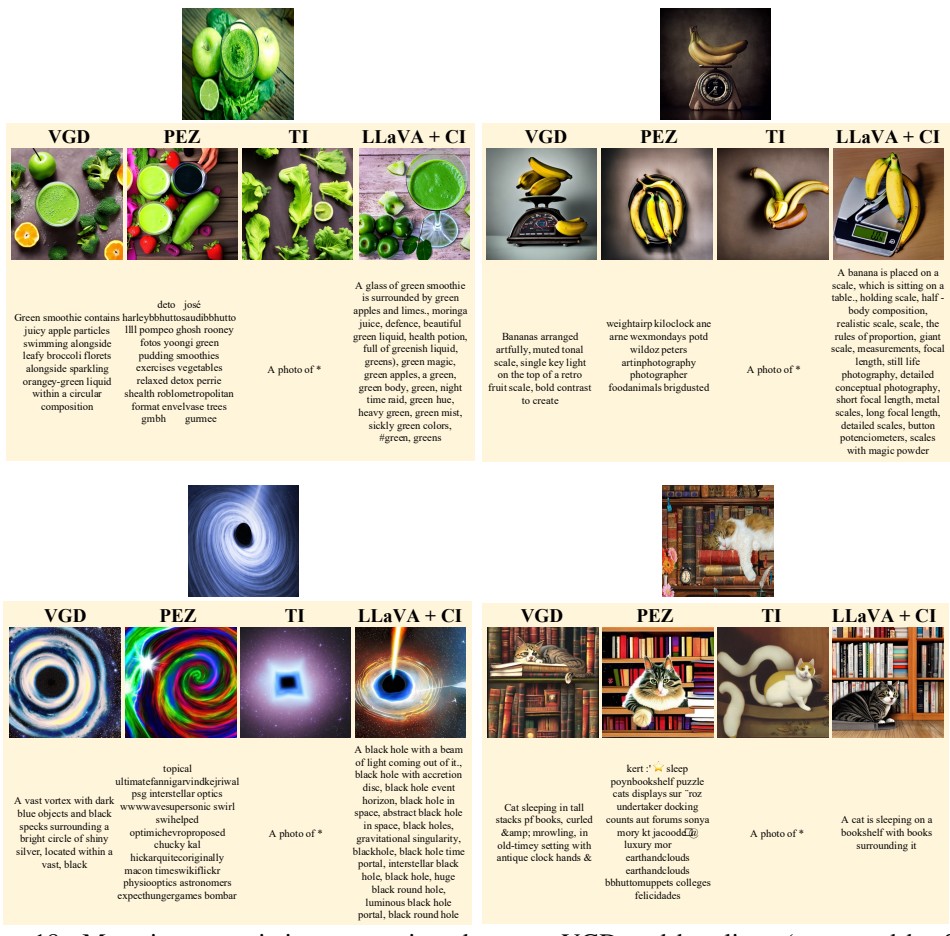

Figure 18: More image variation comparison between VGD and baselines (generated by Stable Diffusion).

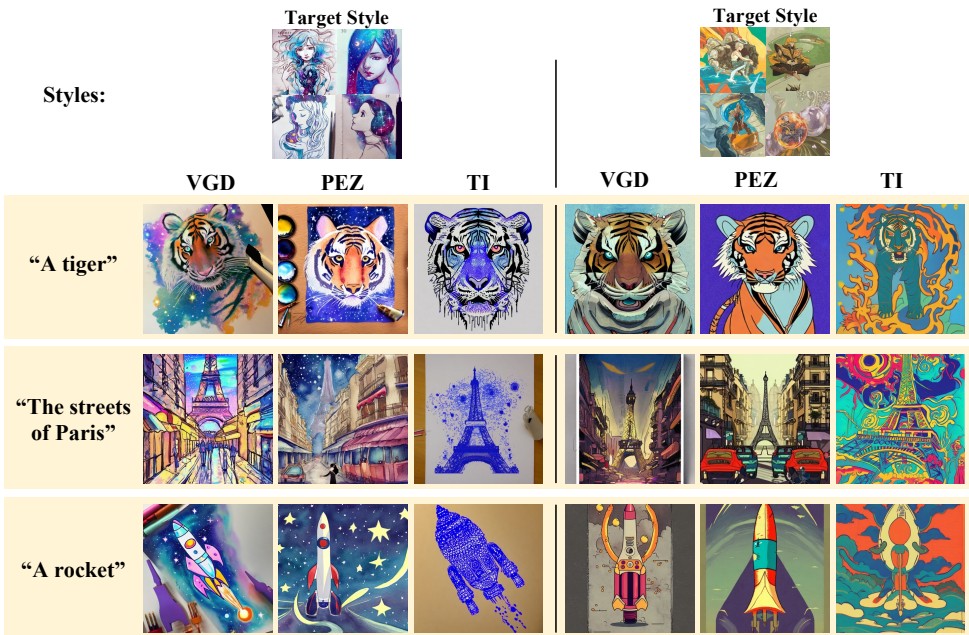

Figure 19: Style transfer comparison between VGD and baselines.

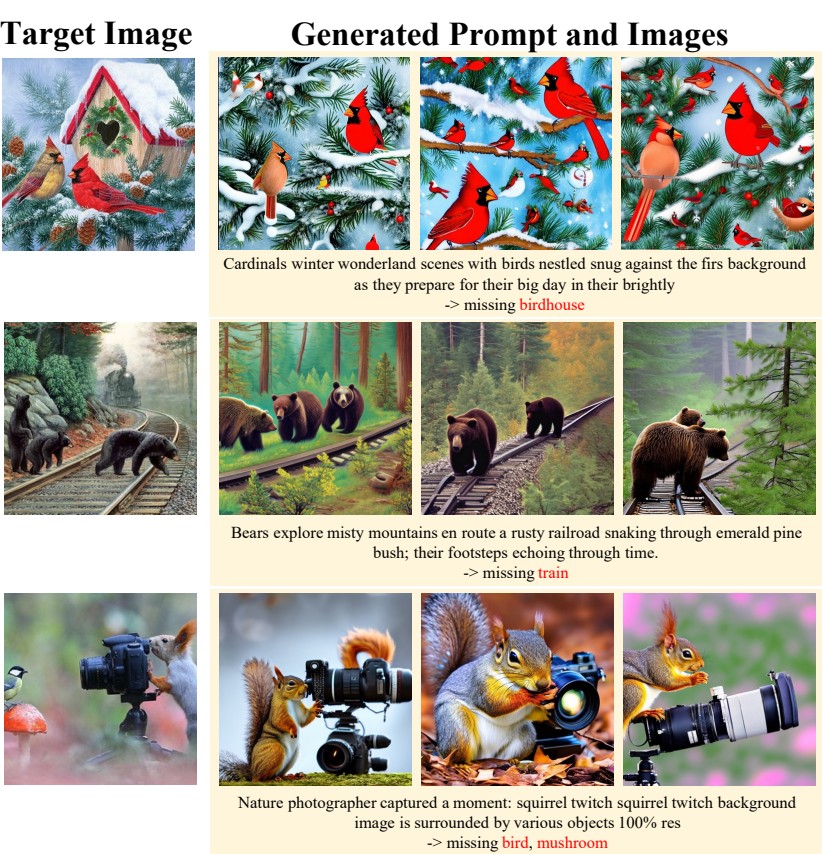

Figure 20: Bad case examples of VGD.

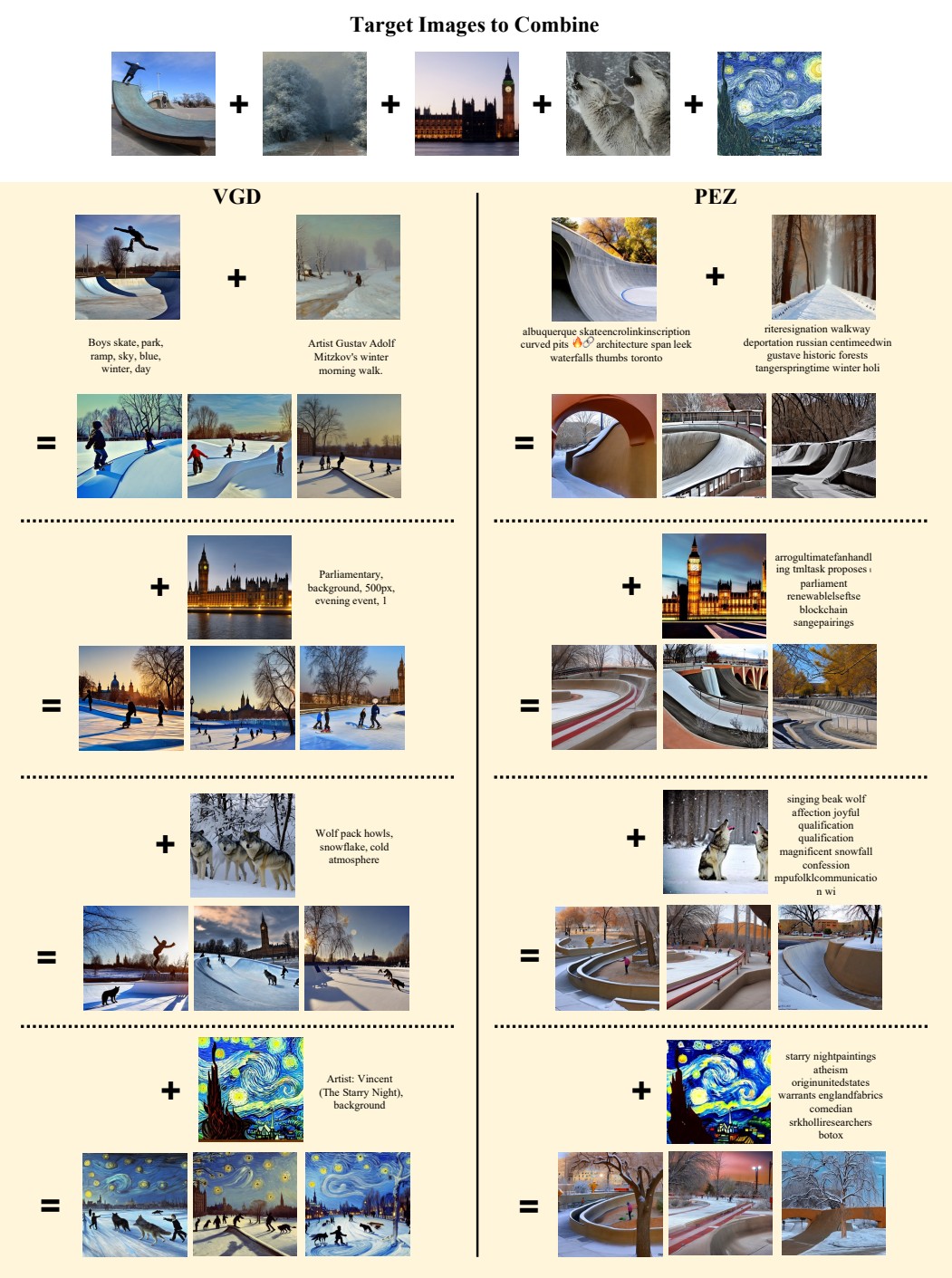

Figure 21: Five-concept image generation comparison between VGD and PEZ.

