# OpenReview forum: "Visually Guided Decoding: Gradient-Free Hard Prompt Inversion with Language Models"
_ICLR.cc/2025/Conference — ICLR 2025 Poster_

### Official Review · Reviewer_CweT · 2024-11-03

**Soundness:** 2
**Presentation:** 3
**Contribution:** 2
**Rating:** 6
**Confidence:** 5

**Summary:**

The paper introduces Visually Guided Decoding (VGD), a gradient-free approach that leverages large language models (LLMs) and CLIP based guidance to generate coherent and semantically aligned prompts. VGD further uses LLMs to produce human-readable prompts while employing CLIP scores to ensure alignment with user-specified visual concepts. Experiments demonstrate that VGD outperforms existing prompt inversion techniques in generating understandable and contextually relevant prompts.

**Strengths:**

- Nice written paper
- The proposed approach is gradient-free which can save time.
- Results in Fig 7 are good.

**Weaknesses:**

- Comparisons are missing with existing relevant methods which improve T2I generation by refining prompts  such as [1] and [2].
- The score in Table are not consistent as LLaVA 1.5 + CLIP Interrogator is outperforming baseline in many cases.
- CLIP can overly large similarity scores as is proved by many existing works. To get a sense of fine-grained similarity, other metrics should be tried.
- Authors have shown that when different CLIP model is used, the approximation no longer holds. This is obvious because Diffusion model uses CLIP-ViT-B/16 which is aligned with Diffusion latent space, Some other CLIP model will not be in the same space and will ultimately not give good results. While this ablation is good, however, this is not novel and not necessary.
- Overall, the paper is good, however, there are some concerns regarding the quantitative evaluations, missing comparisons with other existing works etc.





References

[1] Hao, Yaru, et al. "Optimizing prompts for text-to-image generation." NeurIPS 2024.

[2] Zhan, J., Ai et al. Prompt Refinement with Image Pivot for Text-to-Image Generation. ACL 2024.

**Questions:**

- Are baselines also evaluated across 5 runs like the proposed method?

---

> ### Author Response · Authors · 2024-11-22
> **Response to Reviewer CweT**
>
> ***1. Comparisons are missing with existing relevant methods which improve T2I generation by refining prompts such as [1] and [2].***
>
> - [1] Hao, Yaru, et al. "Optimizing Prompts for Text-to-Image Generation." NeurIPS 2024.
> - [2] Zhan, J., Ai et al. "Prompt Refinement with Image Pivot for Text-to-Image Generation." ACL 2024.
>
> **Response:**
> Please note that the suggested references [1,2] are not relevant to our approach due to the fundamental differences in objectives and methodology. In fact, the main theme of [1,2] is to refine user-provided text prompts to enhance image quality by adapting text prompts to align with model-specific preferences. In contrast, VGD generates prompts directly from images (image-to-prompt inversion). For this reason, it is very difficult to make a fair comparison between VGD and suggested methods [1,2].
>
> ***2. The scores in the table are not consistent as LLaVA 1.5 + CLIP Interrogator is outperforming the baseline in many cases.***
>
> **Response:**
> We thank the reviewer’s comment. Per reviewer’s request, we carefully double-checked our simulation results. First of all, we would like to clarify that the "VGD" results in Table 1 refer to LLaMA2 + VGD. These results represent the performance of our proposed method when combined with the LLaMA2 language model. We observe that LLaVA 1.5 + CLIP Interrogator outperforms LLaMA2 + VGD in some cases, this is because it leverages LLaVA, a model specifically trained on multimodal tasks with image-related data.
>
> However, when LLaVA is combined with our method (LLaVA + VGD), the image quality is on par or surpasses all other approaches across every metric. This is evident in Table 1 and highlights the robustness and adaptability of our method, even when paired with a strong multimodal model like LLaVA.
>
> Moreover, it is noteworthy that LLaMA2 + VGD achieves results slightly lower than those obtained with multimodal models, despite relying solely on a language model that has not been trained on images. This demonstrates the strength of our method, as it bridges the gap between unimodal and multimodal models and ensures high-quality performance in generating interpretable prompts.
>
> ***3. CLIP can produce overly large similarity scores as proven by many existing works. To get a sense of fine-grained similarity, other metrics should be tried.***
>
> **Response:**
> As suggested, we have added kernel inception distance (KID) measurements to quantify the quality and diversity of the generated images (see Table below). We also conducted human evaluations to image variation and style transfer quality.
>
> |                     | #Tokens | LAION-400M    | MS COCO       | Celeb-A       | Lexica.art    |
> |---------------------|---------|---------------|---------------|---------------|---------------|
> | **VGD**             | 32      | 22.71±1.91    | 3.18±1.05     | 72.89±5.10    | 4.32±1.57     |
> | **VGD+LLaVA**       | 32      | 20.14±1.68    | 2.42±0.86     | 54.73±3.58    | 3.86±1.39     |
> | **VGD+LLaVA**       | 77      | 20.07±1.96    | 2.50±1.00     | 54.56±3.86    | 3.29±1.30     |
> | **PEZ**             | 32      | 20.44±2.39    | 4.61±1.21     | 44.68±3.27    | 4.23±1.47     |
> | **LLaVA-1.5 + CI**  | ~77     | 21.48±1.97    | 5.59±1.68     | 60.96±4.16    | 3.13±1.06     |
> | **Textual Inversion** |    1     | 19.89±1.94    | 3.07±1.05     | 55.17±4.50    | 3.74±1.35     |
> | **PH2P**            |    16     | -             | -             | -             | 3.05±1.16     |
>
>
> ***4. Authors have shown that when a different CLIP model is used, the approximation no longer holds. This is obvious because the Diffusion model uses CLIP-ViT-B/16 which is aligned with the Diffusion latent space. Some other CLIP models will not be in the same space and will ultimately not give good results. While this ablation is good, however, this is not novel and not necessary.***
>
> **Response:**
> We agree with the reviewer’s observation that using a different CLIP model leads to a decline in performance. While this outcome might seem expected, we conducted this ablation study to confirm our theoretical understanding and validate the approximation. By including this experiment, we aimed to demonstrate the robustness of our approach and clarify the cases when the alignment assumption is violated.

---

> ### Author Response · Authors · 2024-11-22
> **Response to Reviewer CweT**
>
> ***5. Overall, the paper is good; however, there are some concerns regarding the quantitative evaluations and missing comparisons with other existing works.***
>
> **Response:**
> Thank you for your positive feedback on our work. To address your concerns on quantitative evaluations and missing comparisons, we have conducted additional experiments by comparing our method with the soft-prompt technique Textual Inversion and the hard-prompt technique PH2P (see Table below). We have also updated Fig. 13, 14, 15, 16 and 17 in Appendix to include comprehensive comparisons with these baseline approaches.
>
>
> | Method                | #Tokens | CLIP-Score | Precision | Recall    | F1     |
> |-----------------------|---------|------------|-----------|-----------|--------|
> | **Textual Inversion** | 1       | 0.597      | -         | -         | -      |
> | **LLaVA-1.5**         | 32      | 0.580      | 0.858     | 0.780     | 0.817  |
> | **LLaVA-1.5 + CI**    | ~77     | 0.762      | 0.818     | 0.819     | 0.819  |
> | **LLaVA-1.5 + VGD**   | 32      | 0.769      | 0.835     | 0.793     | 0.813  |
> |                       | ~77     | 0.785      | 0.810     | 0.794     | 0.802  |
> | **PH2P**              | 16      | 0.678      | 0.805     | 0.780     | 0.792  |
> | **PEZ**               | 16      | 0.743      | 0.772     | 0.783     | 0.777  |
> |                       | 32      | 0.745      | 0.752     | 0.784     | 0.768  |
> |                       | 64      | 0.728      | 0.734     | 0.783     | 0.758  |
> | **VGD**               | 16      | 0.700      | 0.827     | 0.779     | 0.802  |
> |                       | 32      | 0.735      | 0.816     | 0.786     | 0.801  |
> |                       | 64      | 0.753      | 0.799     | 0.789     | 0.794  |
> |                       | ~77     | 0.754      | 0.801     | 0.791     | 0.795  |
>
>
> ***6. Are baselines also evaluated across 5 runs like the proposed method?***
>
> **Response:**
> Yes, all baseline methods were evaluated across the same 5 runs to ensure consistency and fairness.

---

> > ### Comment · Reviewer_CweT · 2024-11-25
> >
> > I appreciate the authors for detailed response. While some of my concerns have been answered. I still see some inconsistencies in the above results posted by authors. VGD doesn't seem to be highly effective in all the cases which raises questions on its adaptability across other models. Hence I believe paper needs more work.

---

> > > ### Author Response · Authors · 2024-11-25
> > > **Response to the Reviewer CweT**
> > >
> > > Thank you for your feedback. We would like to clarify your concern regarding the "adaptability across other models." Could you please elaborate on what specific aspects of adaptability you are referring to?

---

> ### Comment · Reviewer_CweT · 2024-11-26
>
> The VGD integration is only shown with LLaVA while there are lot of other multi-modal models. Hence, in current settings the adaptability of VGD across models is not clear. Further  LLaVA-1.5 + CI (77 tokens) has 0.819 F1 score while LLaVA-1.5 + VGD (77 tokens) has 0.802. There are clearly inconsistencies. There are similar cases in paper (Table1)

---

> ### Author Response · Authors · 2024-11-26
> **Response to Reviewer CweT**
>
> We thank the reviewer for the feedback.
>
> **1. Adaptability across the model**
>
> We appreciate the reviewer’s interest in understanding the adaptability of VGD across different models. The current paper demonstrates that VGD integrated with LLaVA to showcase its effectiveness in utilizing vision-language models. While the primary focus of VGD is on leveraging language models (LLMs), we included one language-vision model (LMM), LLaVA, as an example to illustrate the concept. However, to address concerns about adaptability, we have conducted ablation experiments with other language models (Mistral, LLaMA 2, and LLaMA 3), and the results confirm that VGD is effective across a wide range of models (see Table 4). These experiments demonstrate that VGD is not restricted to a specific LLM or LMM and is adaptable across different foundation models.
>
> Table. 4: Ablation on language models.
> | LLM            | #Tokens | LAION CLIP-I | MS COCO Precision | MS COCO Recall | MS COCO F1 | Celeb-A Precision | Celeb-A Recall | Celeb-A F1 | Lexica.art Precision | Lexica.art Recall | Lexica.art F1 |
> |----------------|---------|--------------|-------------------|----------------|------------|-------------------|-----------------|--------------|----------------------|-------------------|----------------|
> | **Mistral-7B** | 16      | 0.443        | 0.615             | 0.422          | 0.706      | 0.808             | 0.850           | 0.828        | 0.815                | 0.780             | 0.797          |
> |                | 32      | 0.463        | 0.656             | 0.466          | 0.729      | 0.803             | 0.858           | 0.829        | 0.808                | 0.787             | 0.797          |
> |                | 64      | 0.497        | 0.680             | 0.476          | 0.751      | 0.783             | 0.858           | 0.818        | 0.791                | 0.789             | 0.790          |
> | **LLaMA2-7B**  | 16      | 0.484        | 0.650             | 0.482          | 0.700      | 0.833             | 0.862           | 0.847        | 0.827                | 0.779             | 0.802          |
> |                | 32      | 0.493        | 0.670             | 0.506          | 0.735      | 0.818             | 0.868           | 0.842        | 0.816                | 0.786             | 0.801          |
> |                | 64      | 0.511        | 0.678             | 0.514          | 0.753      | 0.787             | 0.863           | 0.823        | 0.799                | 0.789             | 0.794          |
> | **LLaMA3-8B**  | 16      | 0.480        | 0.648             | 0.521          | 0.731      | 0.820             | 0.861           | 0.840        | 0.829                | 0.784             | 0.806          |
> |                | 32      | 0.506        | 0.661             | 0.540          | 0.752      | 0.805             | 0.864           | 0.833        | 0.817                | 0.790             | 0.803          |
> |                | 64      | 0.515        | 0.685             | 0.543          | 0.769      | 0.769             | 0.858           | 0.811        | 0.796                | 0.794             | 0.795          |
>
>
> **2. Metrics in Table 1**
>
> We acknowledge that LLaVA + CI (77 tokens) shows higher interpretability as measured by BERTScore, while VGD exhibits slightly lower performance in this metric (0.802 vs. 0.819). However, it is important to emphasize that VGD consistently outperforms LLaVA + CI in the CLIP-I metric for all datasets. This aligns with the primary goal of VGD: generating prompts that are optimized for text-to-image tasks rather than purely for text-based interpretability.
>
> While LLaVA + CI may achieve higher BERTScore, they often fail to align well with the T2I model’s distribution, resulting in lower image quality. In contrast, VGD ensures that generated prompts are well-suited for image generation tasks, as evidenced by its superior performance in human evaluation (see Fig.11). If the main objective were purely text-to-text interpretability, using a dedicated image captioning model would indeed be preferable; however, for high-quality text-to-image generation, VGD provides the most effective and fastest solution (see Table 7), with a slight trade-off in interpretability.
>
> Table. 7: Prompt generation time of VGD, PEZ, PH2P, Textual Inversion, and CLIP Interrogator.
> | **Method**               | **PEZ** | **PH2P**   | **Textual Inversion** |**CLIP Interrogator**| **VGD**  |
> |--------------------------|-----------|--------------|-----------------|-----------------------|----------|
> | **Prompt Generation Time [Sec]** | 193.14  | 34264.74  | 103.45 |  24.85  | 15.33    |
>
> We hope this addresses the reviewer’s concerns and clarifies the strengths and adaptability of VGD. Please let us know if additional details or further experiments are needed.

---

> ### Author Response · Authors · 2024-11-29
> **New experiment results with BLIP-2**
>
> Dear Reviewer CweT,
>
> In our original experiments, we validated the effectiveness of VGD across several foundation models, including three language models (Mistral, LLaMA 2, LLaMA 3) and one vision-language model (LLaVA-1.5), where VGD showed consistent performance. However, in response to the reviewer’s suggestion, **we also conducted additional experiments with the vision-language model BLIP-2**. The results indicate that VGD generates images of the best quality in BLIP-2, demonstrating its adaptability and effectiveness across different models.
>
> **We would like to emphasize once again that VGD offers the most effective and fastest solution in text-to-image generation, with only a slight trade-off in the interpretability of the caption generated by vision-language model.**
>
> Table: Lexica.art image variation experiment in BLIP-2.
> |  Method      |  #Tokens |  CLIPScore |
> |--------------|----------|------------|
> |    BLIP-2    |    32    |    0.566   |
> |  BLIP-2 + CI |    ~77   |    0.758   |
> | BLIP-2 + VGD |    32    |    0.774   |
> |              |    ~77   |    0.781   |

---

> > ### Comment · Reviewer_CweT · 2024-12-02
> >
> > I appreciate the details analysis of authors. Based on the responses so far, I feel authors have done a lot of work to showcase additional results. Hence, I am increasing my score accordingly.

---

> > > ### Author Response · Authors · 2024-12-02
> > >
> > > Dear Reviewer CweT,
> > >
> > > Thank you for your kind words and for recognizing the additional work we’ve put into the manuscript. We truly appreciate your thoughtful analysis and your decision to increase your score. Your feedback has been invaluable in improving the quality of the paper, and we are grateful for the opportunity to address your concerns.
> > >
> > > Thank you again for your continued support.

---

### Official Review · Reviewer_jnTD · 2024-11-03

**Soundness:** 2
**Presentation:** 3
**Contribution:** 2
**Rating:** 6
**Confidence:** 3

**Summary:**

This paper introduce Visually Guided Decoding (VGD), a gradient method that utilizes LLMs and CLIP-based guidance to generate coherent and semantically aligned texts. The experiment demonstrate that VGD outperforms existing prompt inversion techniques.

**Strengths:**

1.  VGD generates fully interpretable prompts that enhance generalizability across tasks.

2. VGD is a gradient-free method which is more flexible

**Weaknesses:**

1. When apply to more complex open-source models with multiple text encoders like SDXL and SD3, as mentioned  in L220-222, the performance of the method would decline. What's more, when facing non-CLIP based models that utilize T5 as text encoders, the methods is quite limited.

2.The current experimental analysis also appears insufficient.  While the method in the paper shows superior performance compare to previous method, it also should include the experiments about the time and cost for more comprehensive comparison.

**Questions:**

1. When using different model architecture like SDXL and SD3 for image generation, how to deal with the problem mentioned in L220-222.

2. I would appreciate if you could conduct more experimental analysis about comparison between VGD and previous methods.

---

> ### Author Response · Authors · 2024-11-22
> **Response to Reviewer jnTD**
>
> ***1. When applied to more complex open-source models with multiple text encoders like SDXL and SD3, as mentioned in L220-222, the performance of the method would decline. Moreover, when facing non-CLIP based models that utilize T5 as text encoders, the method is quite limited.***
>
> **Response:**
> We appreciate the reviewer’s comment. Please understand that modern text-to-image models, including Midjourney, Stable Diffusion (SD), SDXL, and SD3, incorporate CLIP-based text encoders in their architectures. While models like SDXL and SD3 also utilize language models such as T5, they also exploit the CLIP-based encoder. For instance, SDXL employs two text encoders: one based on CLIP and another on T5. Ablation studies in SD3 have shown that even without the T5 encoder, the model maintains comparable performance [1], which underscores the significance of the CLIP-based encoder in the architecture.
>
> Considering that CLIP-based encoders are used predominantly in most text-to-image models, we believe that the validity and effectiveness of our approach will not be degraded.
>
> [1] ESSER, Patrick, et al. "Scaling Rectified Flow Transformers for High-Resolution Image Synthesis." ICML 2024.
>
> ***2. The current experimental analysis also appears insufficient. While the method in the paper shows superior performance compared to previous methods, it should also include experiments about the time and cost for a more comprehensive comparison.***
>
> **Response:**
> As suggested, we have added the results of prompt generation time with single A100 80GB in Table 7. As shown in the table, the proposed VGD does not require any training steps, so that it runs 7 times faster than Textual Inversion, 12 times faster than PEZ, and 2235 times faster than PH2P.
>
> | **Method**               | **PEZ** | **PH2P**   | **Textual Inversion** | **VGD**  |
> |--------------------------|---------|------------|------------------------|----------|
> | **Prompt Generation Time [Sec]** | 193.14  | 34264.74  | 103.45               | 15.33    |
>
>
> ***3. Conduct more experimental analysis comparing VGD with previous methods.***
>
> **Response:**
> We thank the reviewer for the helpful comment. Per your suggestion, we have made the following additions to our experimental analysis:
>
> 1. **Comparison with More Baseline Techniques (see Fig. 13, and 14)**
>    We have included comparisons with the soft-prompt technique Textual Inversion [1] and the hard-prompt technique PH2P [2] in Table 1. These comparisons demonstrate that our method consistently outperforms baseline approaches in both interpretability and prompt quality.
>
> 2. **Generalization of Generated Prompt (see Fig.15, 16, and 17)**
>    To evaluate the robustness of our generated prompt, we tested the prompts generated for Stable Diffusion (SD) on different models (i.e., MidJourney) without any re-optimization. The results show that VGD can still maintain high-quality image generation capability even when applied to a model it was not explicitly designed for.
>
> 3. **Comprehensive Quality Evaluations (see Fig. 11 and 12)**
>    In order to provide both objective and subjective validation of our approach, we have added kernel inception distance (KID) quantifying the quality and diversity of the generated images. Additionally, we conducted human evaluations to assess image variation and style transfer quality.
>
> [1] GAL, Rinon, et al. "An Image is Worth One Word: Personalizing Text-to-Image Generation Using Textual Inversion." ICLR 2023.
>
> [2] MAHAJAN, Shweta, et al. "Prompting Hard or Hardly Prompting: Prompt Inversion for Text-to-Image Diffusion Models." CVPR 2024.

---

> ### Author Response · Authors · 2024-11-27
> **Dear Reviewer jnTD**
>
> Dear Reviewer jnTD,
>
> Thank you once again for your insightful comments.
>
> As the discussion deadline approaches, we look forward to your follow-up response to our rebuttal comments. We kindly request a fair and thorough evaluation of our efforts.
>
> Please feel free to reach out if you need any additional information or have further questions regarding our paper.
>
> Best regards,
>
> Authors of Submission 1743

---

> > ### Comment · Reviewer_jnTD · 2024-11-28
> > **Response to the author**
> >
> > Thank you for your detailed response.
> > According to the prompt feedback and updated manuscript, the authors have addressed all my concerns and I have raised my score.

---

> > > ### Author Response · Authors · 2024-11-28
> > >
> > > Dear Reviewer jnTD,
> > >
> > > Thank you very much for your thoughtful and positive feedback. We are glad to hear that the updates in the manuscript and our responses have addressed your concerns. We sincerely appreciate your time and effort in reviewing our work, and we're grateful for your updated score. Your constructive comments have been invaluable in improving the quality of the manuscript, and we are grateful for your support.

---

### Official Review · Reviewer_uCXB · 2024-11-06

**Soundness:** 2
**Presentation:** 3
**Contribution:** 3
**Rating:** 6
**Confidence:** 3

**Summary:**

The paper presents a gradient-free method, Visually Guided Decoding, that integrates LLMs and CLIP-based guidance to generate coherent and interpretable prompts for text-to-image generation. Compared to traditional methods, VGD can create readable prompts by using LLMs and optimizing these prompts with visual cues by CLIP scores. Experiments indicate that VGD outperforms existing techniques in prompt quality and interpretability on several datasets.

**Strengths:**

- VGD produces coherent and human-readable prompts, facilitating user interaction and modification.
- The training-free method allows easy integration with different LLMs, enhancing adaptability.
- Demonstrates superior performance in generating contextually relevant prompts, as supported by both qualitative and quantitative results.

**Weaknesses:**

- This paper does not analyze bad cases.
- The evaluation lacks depth, especially in semantic aspect evaluation. No human evaluation was conducted. Since image generation is a complex, semantically rich task, CLIPScore may not fully capture true image-prompt alignment, and its classification granularity is limited. Style transfer also requires human evaluation, but the paper only shows a few examples.
- The paper evaluates only semantics without assessing image quality. There should be a discussion on whether using LLMs to create prompts could cause prompts to deviate from the training distribution, potentially lowering image quality.

**Questions:**

In what scenarios might the method generate inaccurate prompts?

---

> ### Author Response · Authors · 2024-11-22
> **Response to Reviewer uCXB**
>
> ***1. This paper does not analyze bad cases.***
>
> **Response:**
> We appreciate the reviewer’s suggestion. In the Appendix of the revised manuscript, we have included a detailed explanation and various examples of bad cases where our method struggles to capture fine-grained details of regional or background objects in complex images (see Figure 18 in the revised manuscript).
>
> In our investigation, we observe that these cases occur when the image contains multiple objects. This is because VGD has difficulty in generating prompts for regional or local objects in the background. We believe this phenomenon is mainly due to the CLIP vision encoder's tendency to prioritize main objects in an image (see [1, 2, 3, 4, 5]).
>
> [1] JING, Dong, et al. FineCLIP: Self-distilled Region-based CLIP for Better Fine-grained Understanding. NerIPS 2024.
>
> [2] Roni Paiss, et al. Teaching clip to count to ten. ICCV, 2023.
>
> [3] Kanchana Ranasinghe, et al. Perceptual grouping in contrastive vision-language models. ICCV, 2022.
>
> [4] Mert Yuksekgonul, et al. When and why vision-language models behave like bags-of-words, and what to do about it? ICLR 2023.
>
> [5] Yiwu Zhong, et al. Regionclip: Region-based language-image pretraining. CVPR, 2022.
>
> ***2. The evaluation lacks depth, especially in semantic aspect evaluation. No human evaluation was conducted. Since image generation is a complex, semantically rich task, CLIPScore may not fully capture true image-prompt alignment, and its classification granularity is limited. Style transfer also requires human evaluation, but the paper only shows a few examples.***
>
> **Response:**
> We agree with the reviewer’s point. To address the reviewer’s comment, we measured kernel inception distance (KID), a metric used to assess the quality and diversity of generated images. We also conducted a human evaluation to gain insight on the image-prompt alignment. The obtained results in Table~ and~ demonstrate the robustness of VGD in both quantitative and qualitative perspective.
>
> ***3. The paper evaluates only semantics without assessing image quality. There should be a discussion on whether using LLMs to create prompts could cause prompts to deviate from the training distribution, potentially lowering image quality.***
>
> **Response:**
> In our understanding, the reviewer is asking about the potential mismatch between the text distribution in the T2I training dataset and the text distribution generated by LLMs. We would like to point out that VGD actively mitigates this issue. Unlike approaches that rely solely on image captioning models like LLaVA, which often fail to generate prompts aligned with T2I model’s distribution, our method addresses this mismatch by exploiting the T2I model's probability in the text decoding process of language models. By aligning the generated prompts with the T2I model's distribution in the decoding, VGD improves compatibility between the prompts and the T2I model.
>
> |                     | #Tokens | LAION-400M    | MS COCO       | Celeb-A       | Lexica.art    |
> |---------------------|---------|---------------|---------------|---------------|---------------|
> | **VGD**             | 32      | 22.71±1.91    | 3.18±1.05     | 72.89±5.10    | 4.32±1.57     |
> | **VGD+LLaVA**       | 32      | 20.14±1.68    | 2.42±0.86     | 54.73±3.58    | 3.86±1.39     |
> | **VGD+LLaVA**       | 77      | 20.07±1.96    | 2.50±1.00     | 54.56±3.86    | 3.29±1.30     |
> | **PEZ**             | 32      | 20.44±2.39    | 4.61±1.21     | 44.68±3.27    | 4.23±1.47     |
> | **LLaVA-1.5 + CI**  | ~77     | 21.48±1.97    | 5.59±1.68     | 60.96±4.16    | 3.13±1.06     |
> | **Textual Inversion** |    1     | 19.89±1.94    | 3.07±1.05     | 55.17±4.50    | 3.74±1.35     |
> | **PH2P**            |    16     | -             | -             | -             | 3.05±1.16     |

---

> > ### Comment · Reviewer_uCXB · 2024-11-26
> >
> > Thank you for the author's feedback. Could the author please specify more details about concern 3.?

---

> > > ### Author Response · Authors · 2024-11-26
> > > **Response to the reviewer uCXB**
> > >
> > > Thank you for your thoughtful follow-up question.
> > >
> > > If we understand your concern correctly, you are suggesting that the text distribution generated by the language model (LLM) could differ from the training text distribution of the T2I model, potentially leading to lower image quality. While this concern might hold for methods relying solely on language models (LLMs or LMMs), we would like to emphasize that VGD actively mitigates this issue through CLIP guidance during the text decoding process.
> > >
> > > The use of CLIP guidance ensures that the generated prompts are aligned not just with the language model’s distribution but also with the T2I model’s expected text-to-image alignment (see Section 3.2. Step 2 - approximation with CLIP score paragraph). This process helps address the potential mismatch between the text distributions, preserving or even enhancing image quality.
> > >
> > > To support this claim, we would like to highlight two key results from the paper:
> > >
> > > **Table 1:**
> > > Images generated by VGD outperform those generated by using pure image captions (e.g., captions generated by LLaVA) in both semantic and visual evaluations. This result demonstrates that VGD’s use of CLIP guidance during prompt generation significantly improves compatibility with the T2I model.
> > >
> > > **Human Evaluation:**
> > > In our human evaluation study, VGD shows much better performance compared to the combination of LLaVA-generated captions and the CLIP Interrogator (LLaVA + CI). This further validates that VGD-generated prompts are more effective at producing high-quality images than prompts generated using only image captioning models.
> > > These results provide clear evidence that VGD effectively bridges the potential gap between the language model’s distribution and the T2I model’s training distribution, ensuring both semantic alignment and high-quality image generation.
> > >
> > > We hope this clarification addresses your concern. Please let us know if further details or experiments are required.

---

> > > > ### Author Response · Authors · 2024-11-29
> > > > **Response to the reviewer uCXB**
> > > >
> > > > Dear Reviewer uCXB,
> > > >
> > > > Thank you once again for your insightful comments.
> > > >
> > > > As the discussion deadline approaches, we look forward to your follow-up response to our rebuttal comments. We kindly request a fair and thorough evaluation of our efforts.
> > > >
> > > > Please feel free to reach out if you need any additional information or have further questions regarding our paper.
> > > >
> > > > Best regards,
> > > >
> > > > Authors of Submission 1743

---

> > > > > ### Author Response · Authors · 2024-12-03
> > > > >
> > > > > Dear Reviewer uCXB
> > > > >
> > > > > As only 10 hours remain for discussion, we would greatly appreciate it if you could let us know if any aspects of our feedback are unclear or if there are any additional points we need to address.
> > > > >
> > > > > We are eager to ensure all concerns are fully resolved and welcome any further guidance you can provide. In addition, we kindly request a re-evaluation of our revisions based on the updates we’ve made in response to your previous comments.
> > > > >
> > > > > Thank you for your time and consideration, and we look forward to your insights.

---

### Official Review · Reviewer_tBYJ · 2024-11-06

**Soundness:** 3
**Presentation:** 3
**Contribution:** 2
**Rating:** 6
**Confidence:** 4

**Summary:**

In this paper, the authors tackle the problem of hard prompt inversion for text-to-image generative models, which is to find a piece of text input to these text-to-image models that can yield images that consist of similar visual concepts in some reference images. Unlike prior methods, which generate soft tokens or completely incomprehensible prompts, the authors propose to combine the language priors in LLMs and CLIP similarity objective to generate human-readable prompts. The authors provide some qualitative and quantitative experimental evidence to demonstrate their claims on the tasks of single image prompt inversion and multi-image concept combination.

**Strengths:**

The authors propose a pretty creative way to conduct gradient-free text-to-image prompt inversion and incorporate language priors in the process. The qualitative results also show some obvious improvement on CLIP-I scores when used with Llava.

**Weaknesses:**

My main concern about this paper lies in the experiments. In general, I am not very convinced by their result that this method significantly improves upon the existing literature.
1. The authors mainly conduct qualitative comparison with PEZ and textual inversion, and not with CLIP-Interrogator, which is very misleading given that CLIP-Interrogator is the best performing baseline based on Table 1 and it can also generate prompts that have similar human interpretability in comparison to the proposed method when used with Llava or BLIP.
2. The authors mention PH2P in their literature review but did not use it as a baseline. From the PH2P paper it seems that they can also obtain similar human interpretability.
3. PEZ also has a variation that incorporates language fluency objectives (Section 5 in PEZ paper). Since this is the main contribution of this paper, the authors should consider comparing it with this variation too.
4. Authors should also consider (at least conceptually) compare with PRISM (https://arxiv.org/pdf/2403.19103), which is another prompt inversion method that uses VLM in their process and can achieve pretty good human interpretability.
5. The main contribution of this paper is to generate human readable prompts. However, the authors fail to provide a principle and quantitative way to measure this contribution. Metrics like perplexity can be easily implemented here.
6. The authors only compare performance on one text-to-image model and fail to demonstrate the generalizability of the inverted prompts on different text-to-image models.
7. The qualitative comparisons provided in this paper are very limited.
8. No limitation section or ethic statement. Given the potential malicious usage of this method (e.g. to generate inappropriate contents), I would encourage the authors to include these sections.

**Questions:**

It would be great if the authors can address the weakness mentioned above.

---

> ### Author Response · Authors · 2024-11-22
> **Response to Reviewer tBYJ**
>
> ***1. The authors mainly conduct a qualitative comparison with PEZ and textual inversion, and not with CLIP-Interrogator, which is very misleading given that CLIP-Interrogator is the best performing baseline based on Table 1 and it can also generate prompts that have similar human interpretability in comparison to the proposed method when used with Llava or BLIP.***
>
> **Response:**
> We appreciate the reviewer’s comment. As suggested, we have added the CLIP-Interrogator's results in the revised manuscript. As shown in Fig. 16 and 17  of the revised manuscript, CLIP-Interrogator’s dependence on a fixed keyword bank limits its ability to handle objects not being represented in the pre-collected 100K keyword bank. We also would like to mention that to evaluate 100K keywords through CLIP encoder will cause a serious computational burden (see Table 7). Since the proposed VGD does not rely on a predefined keyword bank or exhaustive trial-and-error process, it offers greater adaptability and computational efficiency in generating high-quality prompts. Please note that the interpretability of the generated prompts has already been quantified using BERTScore metrics in Table 1, from which we can observe the higher interpretability of VGD over the CLIP-Interrogator.
>
> While CLIP Interrogator captures objects effectively by generating captions using LLaVA and predefined rich keyword bank, there are some critical weaknesses in prompts generated by the CLIP Interrogator (see Appendix Section C.3).
>
> | Method                | #Tokens | CLIP-Score | Precision | Recall    | F1     |
> |-----------------------|---------|------------|-----------|-----------|--------|
> | **Textual Inversion** | 1       | 0.597      | -         | -         | -      |
> | **LLaVA-1.5**         | 32      | 0.580      | 0.858     | 0.780     | 0.817  |
> | **LLaVA-1.5 + CI**    | ~77     | 0.762      | 0.818     | 0.819     | 0.819  |
> | **LLaVA-1.5 + VGD**   | 32      | 0.769      | 0.835     | 0.793     | 0.813  |
> |                       | ~77     | 0.785      | 0.810     | 0.794     | 0.802  |
> | **PH2P**              | 16       | 0.678      | 0.805     | 0.780     | 0.792  |
> | **PEZ**               | 16      | 0.743      | 0.772     | 0.783     | 0.777  |
> |                       | 32      | 0.745      | 0.752     | 0.784     | 0.768  |
> |                       | 64      | 0.728      | 0.734     | 0.783     | 0.758  |
> | **VGD**               | 16      | 0.700      | 0.827     | 0.779     | 0.802  |
> |                       | 32      | 0.735      | 0.816     | 0.786     | 0.801  |
> |                       | 64      | 0.753      | 0.799     | 0.789     | 0.794  |
> |                       | ~77     | 0.754      | 0.801     | 0.791     | 0.795  |
>
>
> ***2. The authors mention PH2P in their literature review but did not use it as a baseline. From the PH2P paper it seems that they can also obtain similar human interpretability.***
>
> **Response:**
> In the revised manuscript, we have added the quantitative and qualitative results of PH2P. As can be seen from Table below, VGD outperforms PH2P in terms of BERTScore and CLIP-I score. Furthermore , we observe images generated by VGD are more semantically aligned with the original ones than those generated by PH2P.
>
> Table: image quality (CLIP-I score) and prompt quality (BERTScore) comparison in Lexica.art.
>
>
> ***3. PEZ also has a variation that incorporates language fluency objectives (Section 5 in PEZ paper). Since this is the main contribution of this paper, the authors should consider comparing it with this variation too.***
>
> **Response:**
> We thank the reviewer’s comment. Please note that the language fluency objective [1] in Section 5 of PEZ is unrelated to VGD. Section 5 of PEZ paper discusses the text-to-text generation issue of the language model hard prompting. The fluency objective, designed for enhancing language model capability of text-based applications, works for the text-to-text generation tasks (see Original FluentPrompt [1]). Noting that the main focus of VGD is the text-to-image prompt inversion, the objective of our work is clearly distinct from the fluency objective. We are willing to provide more discussion per the reviewer’s request.
>
> [1] Shi, Weijia, et al. Toward Human Readable Prompt Tuning: Kubrick's The Shining is a good movie, and a good prompt too?. EMNLP2023.

---

> ### Author Response · Authors · 2024-11-22
> **Response to Reviewer tBYJ**
>
> ***4. Authors should also consider (at least conceptually) compare with PRISM ([https://arxiv.org/pdf/2403.19103](https://arxiv.org/pdf/2403.19103)), which is another prompt inversion method that uses VLM in their process and can achieve pretty good human interpretability.***
>
> **Response:**
> Please note that the referred work, uploaded to arXiv in March 2024, is currently under review at ICLR 2025 (https://openreview.net/forum?id=hIKsem01M5). Given its status and the lack of open-source code, it is very difficult to perform an apple-to-apple comparison in a short rebuttal period. Nevertheless, in our revised manuscript, we briefly mentioned that PRISM employs an iterative algorithm based on in-context learning of large language models. In a nutshell, it generates image captions, evaluates the generated images through the caption, and then iteratively refines the caption based on evaluation scores. Since, such iterative processes is unnecessary, VGD is much faster and require for less computational burden (see Table 7 in the Appendix of the revised paper).
>
> ***5. The main contribution of this paper is to generate human-readable prompts. However, the authors fail to provide a principle and quantitative way to measure this contribution. Metrics like perplexity can be easily implemented here.***
>
> **Response:**
>  In our work, we measured the fluency of the prompt using BERTScore [1]. Please note that BERTScore, represented by the precision, recall, and F1 score, effectively captures the semantic similarity and interpretability of the prompts [1, 2]. Perplexity, a metric measuring the syntactic fluency rather than semantic quality [3, 4], was rather used in language modeling.
> Recent advancements in text evaluation have shown that metrics like BERTScore are better aligned with human judgments [1], which is why we opted for this metric.
>
> [1] Zhang, Tianyi, et al. Bertscore: Evaluating text generation with bert, ICLR 2020.
>
> [2] MAHAJAN, Shweta, et al. Prompting Hard or Hardly Prompting: Prompt Inversion for Text-to-Image Diffusion Models, CVPR 2024.
>
> [3] Muhlgay, Dor, et al. Generating benchmarks for factuality evaluation of language models. EACL 2024.
>
> [4] HU, Yutong, et al. Can Perplexity Reflect Large Language Model's Ability in Long Text Understanding?. ICLR 2024.
>
> ***6. The authors only compare performance on one text-to-image model and fail to demonstrate the generalizability of the inverted prompts on different text-to-image models.***
>
> **Response:**
> To address the concern on the generalizability, we have included additional results demonstrating how prompts generated by our method and other baseline methods (optimized for the Stable Diffusion (SD) model) behave when applied to a different text-to-image model (see Fig. 13, 14, and 15).
>
> In this experiment, we did not generate any new prompt for the newly tested T2I model (i.e., MidJourney, DALL-E 2), meaning that we simply reused the prompts generated for SD in evaluating the effectiveness of producing high-quality images with various T2I models. In doing so, we could ensure a fair comparison and also highlight the generalizability of the prompts across models.
>
> Finally, we would like to point out that this type of generalization cannot be done with soft prompt methods like Textual Inversion, as they are inherently tied to the specific model they are trained on. When compared to hard prompt inversion methods (such as PH2P and PEZ), VGD generates consistently better results in various text-to-image models, which underscores the robustness and flexibility of the proposed VGD.
>
>
> ***7. The qualitative comparisons provided in this paper are very limited.***
>
> **Response:**
> We thank the reviewer’s comment. In order to provide a more comprehensive evaluation, we have added qualitative results of image variation and style transfer from Textual Inversion and PEZ (see Fig. 16 and 17 of the revised manuscript), which demonstrate the effectiveness of VGD over existing baseline techniques.

---

> > ### Author Response · Authors · 2024-11-22
> > **Response to Reviewer tBYJ**
> >
> > ***8. No limitation section or ethic statement. Given the potential malicious usage of this method (e.g., to generate inappropriate contents), I would encourage the authors to include these sections.***
> >
> > **Response:**
> >  We appreciate the reviewer's suggestion. We have included limitations and ethics statements in the Appendix of the revised manuscript.
> >
> > #### **Limitations**
> > Although we showcase the generalizability of VGD-generated prompts across multiple text-to-image models, our evaluations primarily focus on popular models such as Stable Diffusion and MidJourney. However, since many other models are trained on similar datasets and follow comparable methods and architectures, we anticipate that the results would be consistent across these models as well.
> >
> > #### **Ethics Statement**
> > The development of VGD raises important ethical considerations, particularly regarding potential misuse. While our method improves the interpretability and generalizability of text-to-image generation, it could also be exploited to generate harmful or inappropriate content, such as deepfakes or offensive imagery. We strongly discourage such uses and emphasize that this method is intended for responsible applications in creative, educational, and professional contexts.

---

> > ### Comment · Reviewer_tBYJ · 2024-11-22
> > **Response to authors' rebuttal**
> >
> > Thank you for your rebuttal. This rebuttal has answer weakness 1,2,4,6,8 that I mentioned previously. However, I find my concerns unanswered for weakness 3,5,7.
> >
> > (3) While in the original PEZ paper the fluency objective is mainly tested on text-to-text generation, the same formulation can be directly applied to text-to-image setting as well. Since the main contribution of this paper is to add language prior, completely ignoring this comparison in the paper seems unjustified.
> >
> > (5) As the authors have mentioned, BertScore and perplexity measure different aspects of a language prompt, as a result, I will think it would be good to compare perplexity
> >
> > (7) The "additional" qualitative comparisons the authors provide seem to overlap quite a lot with the qualitative examples they have already included previously. Therefore it does not address my concern.
> >
> > Since the authors have answered more than half of my concern, I am willing to update my score from 3 to 5. However, I am still not in favor of accepting this paper.

---

> > > ### Author Response · Authors · 2024-11-25
> > > **Response to reviewer tBYJ**
> > >
> > > We sincerely thank the reviewer for acknowledging our efforts in addressing the concerns and updating the score from 3 to 5. We deeply appreciate your constructive feedback and are committed to improving the manuscript by addressing the remaining points.
> > >
> > > ***3. PEZ fluency objective.***
> > >
> > > **Response:**
> > > We appreciate the reviewer’s comment. We would like to clarify that the fluency objective (i.e., optimize with negative log-likelihood of text prompt) in the PEZ paper is formulated specifically for text-to-text generation (no experiments are demonstrating its applicability to text-to-image (T2I) tasks in PEZ paper), **as its implementation within a text-to-image (T2I) framework is infeasible**. First, in the case of the CLIP text encoder, the absence of a projection layer (i.e., text projection head) to produce a text distribution makes it fundamentally impossible to integrate a fluency objective into the T2I pipeline. Second, utilizing an external language model to compute the negative log-likelihood of the text introduces critical challenges with the detokenization and tokenization processes. Specifically, the detokenization step (converting SD token indices to text) followed by re-tokenization (via the LLM tokenizer) breaks the gradient path, rendering backpropagation impossible.
> > >
> > > ***5. Perplexity***
> > >
> > > **Response:**
> > > As the reviewer's suggestion, we included a perplexity analysis in the revised manuscript. The results show that our prompts achieve superior perplexity scores, further supporting the quality of our method in generating coherent and interpretable prompts.
> > >
> > > |                | #Tokens | LAION-400M | MS COCO | Celeb-A | Lexica.art |
> > > |:--------------:|:-------:|:----------:|:-------:|:-------:|:----------:|
> > > |       **VGD**      |    32   |    94.96   |  89.48  |  68.69  |    91.12   |
> > > |   **VGD+LLaVA**   |    32   |   190.37   |  106.84 |  108.80 |   121.11   |
> > > |   **VGD+LLaVA**   |    77   |    42.00   |  36.11  |  34.39  |    47.07   |
> > > |       PEZ      |    32   |   813.27   |  766.81 |  971.35 |   659.39   |
> > > | LLaVA-1.5 + CI |   ~77   |   102.62   |  80.97  |  67.59  |    76.04   |
> > > |      PH2P      |    16   |      -     |    -    |    -    |   1126.11  |
> > >
> > > ***7. More qualitative examples***
> > >
> > > **Response:**
> > > We understand the reviewer’s concern regarding the overlap in qualitative comparisons. The original examples were intentionally selected to align with those in the main text to facilitate direct comparisons with the results of our proposed method. However, in response to the reviewer’s request, we have added new qualitative examples using a broader range of images from the web. These additional examples ensure a more diverse and representative evaluation of our approach and further highlight the strengths of our method over existing baselines.
> > >
> > > We hope these updates address the remaining concerns and further demonstrate the robustness and contribution of our work. Thank you again for your valuable insights and for reconsidering your evaluation of our paper.

---

> ### Comment · Reviewer_tBYJ · 2024-11-25
> **Response to authors' comments**
>
> Thank you for your response. However, they do not fully answer my concerns and I would like to elaborate below.
>
> 3. I should clarify my concern regarding this problem: Since in VGD the authors have assumed access to another LLM, if we use the same assumption for PEZ fluency objective for fair comparison, it is reasonable to directly use the CLIP loss as $L_{task}$ and the same $L_{fluency}$ from the LLM as they suggested in PEZ paper. In this case, what is the difference between VGD and PEZ fluency?
>
> 5. Thank you for including this additional experiment. However, the results in this table is quite puzzling. Why do longer prompts have significantly lower perplexity than the shorter prompts, even when tested with the same VGD + LLaVA? Also the fact that PH2P has significantly higher perplexity than PEZ does not align with the PH2P qualitative results (in PH2P paper).
>
> 7. Thank you for including the additional example. They have addressed my concern in this regard.

---

> > ### Author Response · Authors · 2024-11-26
> > **Response to Reviewer tBYJ**
> >
> > Thank the reviewer for the feedback!
> >
> > ***3. Fluency objective***
> >
> > We would like to clarify that applying an external language model to PEZ is technically infeasible.
> >
> > In PEZ, the fluency objective should rely on gradient-based optimization that optimizes the generated prompt using negative log-likelihood (NLL) from the external language model. For this to work, the optimization process requires the ability to backpropagate through the entire pipeline (language model output to prompt embedding of text conditioning network of SD), which includes the tokenization and detokenization steps. This conversion step — from CLIP token indices to LLM token indices — introduces a break in the gradient path because the detokenization (from clip text indices to text) followed by re-tokenization (from text to language model indices) through 2 different tokenizers (i.e., CLIP and Language model) is not differentiable. As a result, it becomes impossible to update the prompt embedding using the fluency loss ($L_{fluency}$) in a way that allows backpropagation through the entire pipeline.
> >
> > On the other hand, in VGD, we utilize an external language model for prompt generation in a gradient-free (training-free) manner. This means that VGD operates without training, the prompt is generated through the text decoding process of LLM with CLIP score guidance.
> >
> > ***4. Perplexity***
> >
> > Regarding the perplexity results, we understand your concern about why longer prompts have significantly lower perplexity than shorter prompts. While this might seem counterintuitive from a purely language-domain perspective, the key factor here is the involvement of the CLIP score during the decoding process.
> >
> > In our experiments, we found that during the decoding process, CLIP score plays a significant role, particularly in the earlier stages of prompt generation. As the prompt length increases, the influence of the CLIP score stabilizes, leading to a decrease in its impact on the perplexity score. The CLIP score has a stronger influence for shorter prompts, as the generated text is more directly shaped by image-related features, resulting in higher perplexity values. As the prompt lengthens, we observe that the CLIP score reaches a point where it no longer significantly improves the quality of the image, and additional tokens selected in the beam search are often text with higher natural language probability, thus reducing perplexity.
> >
> > Essentially, the CLIP score helps guide the generation process, but once the generated prompt has covered the major image-related concepts (e.g., style, content), the remaining tokens are chosen from the most probable natural language phrases, which leads to a lower perplexity for longer prompts.
> >
> > Regarding PH2P and its perplexity results, we would like to clarify that we used the reference code for PEZ (https://github.com/YuxinWenRick/hard-prompts-made-easy) and PH2P (https://github.com/ubc-vision/Prompting-Hard-Hardly-Prompting) as provided by the original authors for our experiments. We have reported the results directly based on their implementations, and the reasons for the discrepancy in perplexity between PEZ and PH2P are unclear to us. Our intention was to conduct the experiments using the existing code as-is, and we have presented the results faithfully without further modifications.

---

> > > ### Comment · Reviewer_tBYJ · 2024-11-26
> > > **Response to the author**
> > >
> > > Thank you for your response.
> > >
> > > 3. Thank you for your response. I think this is a fair explanation. I do want to point out that it is possible to propagate gradients between different tokenizers using https://arxiv.org/pdf/2408.04816. I did not mention this work earlier because it can be considered as a concurrent work. However, what the authors claim to be impossible is in fact possible.
> > >
> > > 7. Again I think this is a fair explanation. However, after examining all the other comments from other reviewers, I have noticed that many of us have questions and concerns about the validity of the experiments. As the last major update I would request from the authors, I would like to suggest the author to include a link to their code in the appendix so that the reviewers can verify the results.
> > >
> > > I think the manuscript has been improved significantly after the rebuttal revisions, therefore I am open to change my score again, but I would like to make this decision after/towards the end of the discussion period after all the discussions are concluded.

---

> ### Author Response · Authors · 2024-11-27
> **Thank the reviewer tBYJ for your constructive comments**
>
> Dear Reviewer tBYJ,
>
> Thank you for your thoughtful feedback. We appreciate your acknowledgment of the improvements made to the manuscript after the rebuttal revisions. We also appreciate your suggestion to include a link to the code for easier verification of our results. Our code is available for download from the supplementary material section on OpenReview, and we hope this helps address any further questions regarding the experiments.
>
> We are grateful for the constructive comments throughout the review process, and we have worked diligently to address all eight concerns raised by the reviewer in the initial rounds. We believe the manuscript has been significantly improved as a result. We kindly ask that you consider this effort in your final evaluation, and we would be grateful if the revised manuscript and its improvements are reflected in the final score.
>
> Thank you once again for your time and valuable feedback.

---

> > ### Comment · Reviewer_tBYJ · 2024-12-03
> >
> > After seeing the discussions the authors had with other reviewers, I believe they did not raise any more concern from my side and I appreciate the efforts the authors put in during the discussion period, therefore I have decided to raise my score from 5 to 6.

---

> ### Author Response · Authors · 2024-12-03
> **Thank you**
>
> Dear Reviewer tBYJ,
>
> Thank you very much for your thoughtful feedback and for taking the time to engage in the discussion process. We greatly appreciate your recognition of our efforts to address the concerns raised and are delighted that the revisions have met your expectations. Your decision to raise your score from 5 to 6 is truly encouraging, and it motivates us to continue striving for the highest quality in our work.
>
> Best regards,
>
> Authors of Submission 1743

---

### Author Response · Authors · 2024-11-22
**Common Response**

We sincerely appreciate the reviewers for their time and effort in reviewing our manuscript and providing valuable comments and insights. We are encouraged by the reviewers' assessment that:

- Our work addresses a crucial challenge in text-to-image generation: creating interpretable prompts for generative models in a gradient-free manner (tBYJ, uCXB, jnTD, CweT).
- Our proposed method, referred to as **Visually Guided Decoding (VGD)**, deliberately combines large language model (LLM) language priors with CLIP conditional probability to generate human-readable prompts (tBYJ, uCXB).
- VGD demonstrates qualitative and quantitative improvements, notably in creating contextually relevant and semantically aligned prompts (uCXB, jnTD, CweT).
- VGD performs comparable to or even better than existing approaches in terms of prompt quality and interpretability (tBYJ, uCXB, jnTD).
- The writing is easy to follow (uCXB), well-written, and concise (CweT).

Again, we thank the reviewers for recognizing the contributions and strengths of our work. We also appreciate the reviewers’ thoughtful comments and useful feedback, mainly on the experimental results. To address the reviewers’ suggestions, we have conducted additional experiments and provided detailed responses to each comment. We hope that our efforts will meet your expectations and further demonstrate the effectiveness of our approach (**Please see Appendix of the revised manuscript.**).

---

> ### Author Response · Authors · 2024-11-25
> **Common Response 2**
>
> Dear Reviewers,
>
> Thank you once again for your thoughtful feedback and for taking the time to evaluate our manuscript. We greatly appreciate the opportunity to address your concerns and have worked diligently to incorporate your suggestions into the revised version of the paper. We would like to highlight a few key updates and clarifications based on your comments:
>
> **1. Prompt generalization with Midjourney and DALL-E 2  (Fig 13, 14, 15 and 16)**
>
> We have conducted additional experiments to test the cross-model applicability of the prompts generated by our method. Specifically, prompts initially generated using Stable Diffusion (SD) were applied to DALL-E 2 and MidJourney to evaluate their generalization ability. This demonstrates the flexibility and robustness of VGD-generated prompts across different text-to-image models, even without model-specific tailoring.
>
> **2. Prompt generation time (Table 7)**
>
> We have measured and reported the computational efficiency of our method in terms of prompt generation time. The results show a significant speed advantage for VGD, which avoids the iterative processes required by competing methods. Specifically, VGD is 7 times faster than Textual Inversion, 12 times faster than PEZ, and 2,235 times faster than PH2P. This efficiency ensures that VGD is highly scalable and practical for real-world applications.
>
> **3. More qualitative comparison (Fig 17, 18 and 19)**
>
> To address concerns about overlap in qualitative examples, we have expanded the range of scenarios and included new qualitative comparisons. These additions cover a broader variety of images, showcasing the adaptability and effectiveness of our method across diverse contexts and applications.
>
> **4. Human evaluation (Fig. 11 and 12)**
>
> We have included a human evaluation study to assess the quality and semantic alignment of generated images. The results indicate that VGD achieves the best performance in image variation (60.5%) and style transfer (52.6%), further validating the effectiveness of our approach.
>
> **5. Perplexity evaluation for interpretability (Table 8)**
>
> In response to reviewer feedback, we have added perplexity evaluation as a complementary metric for interpretability. This metric evaluates the fluency and coherence of the generated prompts. VGD achieves the best interpretability scores, excelling in both BERTScore and perplexity, further demonstrating its ability to produce high-quality, human-readable outputs.
>
> **6. Bad cases (Fig. 20)**
>
> We have included an analysis of failure cases to provide a balanced perspective on the performance of VGD. These examples highlight specific challenges and limitations encountered by our method, offering insights into potential areas for improvement and directions for future work.
>
> We kindly request you review the updated manuscript, as we believe these revisions address the remaining concerns and provide a clearer picture of our contributions. Your feedback has been invaluable in improving the quality and clarity of our work, and we sincerely thank you for your insights.

---

> ### Author Response · Authors · 2024-11-28
> **Dear Reviewers**
>
> Dear Reviewers,
>
> We sincerely thank all the reviewers for your valuable feedback and insightful comments throughout the review process. As a result of the discussions and suggestions, we have significantly improved the manuscript and the experiments. We truly appreciate the constructive critiques, which have helped us refine our work and address all the points raised.
>
> We are also grateful to Reviewer tBYJ, who has already updated their score from 3 to 5, noting that the manuscript has improved significantly after the rebuttal revisions. Furthermore, Reviewer tBYJ has expressed openness to further increasing their score. We greatly appreciate the recognition of the improvements we have made.
>
> We have worked diligently to incorporate additional experiments and revisions in response to the concerns of all reviewers. We are more than happy to engage if any further questions or if additional discussion is needed. However, as the deadline for final decisions approaches, we kindly ask that you take the time to reflect on the improvements made in the revised manuscript when finalizing your evaluations.
>
> Thank you once again for your time and thoughtful consideration. We look forward to hearing from you all.

---

> ### Author Response · Authors · 2024-12-02
> **Dear Reviewers**
>
> Dear Reviewers,
>
> We are grateful for your thoughtful feedback and for the opportunity to revise our manuscript. Following the revisions, we are pleased to share that Reviewer tBYJ has increased the score from 3 to 5, noting that the manuscript has improved significantly after the rebuttal revisions. Furthermore, Reviewer jnTD has raised the score from 5 to 6, stating that their concerns have been fully addressed.
>
> We would like to highlight a few key updates and clarifications based on your comments:
>
> **1. Prompt generalization with Midjourney and DALL-E 2 (Fig 13, 14, 15 and 16)**
>
> We have conducted additional experiments to test the cross-model applicability of the prompts generated by our method. Specifically, prompts initially generated using Stable Diffusion (SD) were applied to DALL-E 2 and MidJourney to evaluate their generalization ability. This demonstrates the flexibility and robustness of VGD-generated prompts across different text-to-image models, even without model-specific tailoring.
>
> **2. Prompt generation time (Table 7)**
>
> We have measured and reported the computational efficiency of our method in terms of prompt generation time. The results show a significant speed advantage for VGD, which avoids the iterative processes required by competing methods. Specifically, VGD is 7 times faster than Textual Inversion, 12 times faster than PEZ, and 2,235 times faster than PH2P. This efficiency ensures that VGD is highly scalable and practical for real-world applications.
>
> **3. More qualitative comparison (Fig 17, 18 and 19)**
>
> To address concerns about overlap in qualitative examples, we have expanded the range of scenarios and included new qualitative comparisons. These additions cover a broader variety of images, showcasing the adaptability and effectiveness of our method across diverse contexts and applications.
>
> **4. Human evaluation (Fig. 11 and 12)**
>
> We have included a human evaluation study to assess the quality and semantic alignment of generated images. The results indicate that VGD achieves the best performance in image variation (60.5%) and style transfer (52.6%), further validating the effectiveness of our approach.
>
> **5. Perplexity evaluation for interpretability (Table 8)**
>
> In response to reviewer feedback, we have added perplexity evaluation as a complementary metric for interpretability. This metric evaluates the fluency and coherence of the generated prompts. VGD achieves the best interpretability scores, excelling in both BERTScore and perplexity, further demonstrating its ability to produce high-quality, human-readable outputs.
>
> **6. Bad cases (Fig. 20)**
>
> We have included an analysis of failure cases to provide a balanced perspective on the performance of VGD. These examples highlight specific challenges and limitations encountered by our method, offering insights into potential areas for improvement and directions for future work.
>
> We kindly request you review the updated manuscript, as we believe these revisions address the remaining concerns and provide a clearer picture of our contributions. Your feedback has been invaluable in improving the quality and clarity of our work, and we sincerely thank you for your insights.
>
> **7. BLIP-2 model evaluation**
>
> We also conducted additional experiments with the vision-language model BLIP-2. The results indicate that VGD generates images of the best quality in BLIP-2, demonstrating its adaptability and effectiveness across different models. We would like to emphasize once again that VGD offers the most effective and fastest solution in text-to-image generation, with only a slight trade-off in the interpretability of the caption generated by vision-language model.
>
>
> Table: Lexica.art image variation experiment in BLIP-2.
> |  Method      |  #Tokens |  CLIPScore |
> |--------------|----------|------------|
> |    BLIP-2    |    32    |    0.566   |
> |  BLIP-2 + CI |    ~77   |    0.758   |
> | BLIP-2 + VGD |    32    |    0.774   |
> |              |    ~77   |    0.781   |
>
>
> We hope the changes made meet your expectations, and if any further questions arise, we are happy to engage in further discussion. As the evaluation deadline approaches, we kindly request that the revised manuscript and our efforts to address all concerns be reflected in the final decision.
>
> Thank you once again for your valuable input.

---

> > ### Author Response · Authors · 2024-12-02
> > **Dear Reviewers**
> >
> > Dear Reviewers,
> >
> > We sincerely appreciate your constructive feedback and the opportunity to improve our manuscript. Following our revisions, we are pleased to report that Reviewer tBYJ has increased their score from **3 to 5**, Reviewer jnTD has raised theirs from **5 to 6**, and Reviewer CweT has also raised their score from **5 to 6**, all indicating that their concerns have been satisfactorily addressed.
> >
> > We hope that the updates made in the revised manuscript meet your expectations. As the evaluation deadline approaches, we kindly ask that the revised manuscript and our efforts to address all concerns be reflected in the final decision.
> >
> > Thank you

---

> > > ### Author Response · Authors · 2024-12-03
> > >
> > > Dear Reviewers,
> > >
> > > We would like to express our sincere gratitude for your constructive feedback and for allowing us to improve our manuscript. We are pleased to report that Reviewer tBYJ has increased the score from **3 to 6**, while Reviewer jnTD has raised the score from **5 to 6**, and Reviewer CweT has also elevated the score from **5 to 6**. These positive changes indicate that the revisions we made have successfully addressed the concerns raised.
> > >
> > > We hope that the updates in the revised manuscript meet your expectations, and we remain open to any further questions or discussions. As the evaluation deadline approaches (5 hours), we kindly request that the revised manuscript and our efforts to address all concerns be considered in the final decision.

---

### Public Comment · ~Donghoon_Kim1 · 2025-07-21

Official code implementation : https://github.com/DonghoonKim-1938/VGD

---

### Meta-Review · Area_Chair_SETT · 2024-12-19

**Metareview:**

This paper addresses the prompt inversion problem, with a gradient-free method that leverages LLMs and CLIP-based guidance to generate relevant prompts for text-to-image generation. The paper was reviewed by four experts in the field who acknowledged that the method proposed is creative (tBYJ) and adaptable / flexible (uCXB, jnTD), produces human-readable prompts which facilitate user interaction (uCXB, jnTD), and that the paper is well written (CweT).

The main concerns raised by the reviewers can be summarized as:
- Experimental validation is not compelling: some baselines are missing (tBYJ, CweT); only one T2I model is considered (tBYJ, jnTD), and so the generalizability of the method remains unclear; qualitative comparisons are not performed against the best performing method (tBYJ); there is no assessment of image quality (uCXB); scores in tables appear inconsistent (CweT); it would be beneficial to consider metrics beyond CLIP similarity scores (CweT)
- It is not clear how to measure in a principled way the impact of the contribution (tBYJ)
- There is no limitations discussion (tBYJ)
- The method appears quite limited beyond CLIP-based encoder (jnTD), and many T2I models use other encoders such as T5.

During rebuttal and discussion period, the authors add results on Dalle and Midjourney, evaluate computational efficiency of the proposed approach, substantially expand qualitative comparisons, they also include a human evaluation user study, add metrics and baselines as suggested by reviewers, and discuss failure cases. There is is quite a bit of back and forth between the authors and the reviewers during the discussion period, which results in reviewers being satisfied with the updated paper and leaning unanimously towards acceptance. The AC acknowledges all the effort put by the authors during the discussion period, and agrees with the reviewers' assessment. Therefore, the AC recommends to accept.

**Additional Comments On Reviewer Discussion:**

No additional comments.

---

### Decision · Program_Chairs · 2025-01-22

Accept (Poster)